Concentrations and radiative forcing of anthropogenic aerosols from 1750-2014 simulated with the OsloCTM3 and CEDS emission inventory

Marianne Tronstad Lund[1*], Gunnar Myhre[1], Amund Søvde Haslerud[1], Ragnhild Bieltvedt Skeie[1], Jan Griesfeller[2], Stephen Matthew Platt[3], Rajesh Kumar[4,5], Cathrine Lund Myhre[3], Michael Schulz[2]

*1 CICERO Center for International Climate Research, Oslo, Norway*

*2 Norwegian Meteorological Institute, Oslo, Norway*

*3 NILU – Norsk institutt for luftforskning, Dept. Atmospheric and Climate Research (ATMOS), Kjeller, Norway*

*4 Advanced Study Program, National Center for Atmospheric Research, Boulder, Colorado, USA*

*5 Atmospheric Chemistry Division, National Center for Atmospheric Research, Boulder, Colorado, USA*

*Corresponding author: m.t.lund@cicero.oslo.no

## Abstract

We document the ability of the new generation Oslo chemistry-transport model, OsloCTM3, to accurately simulate present-day aerosol distributions. The model is then used with the new Community Emission Data System (CEDS) historical emission inventory to provide updated time series of anthropogenic aerosol concentrations and consequent direct radiative forcing (RFari) from 1750 to 2014.

Overall, the OsloCTM3 performs well compared with measurements of surface concentrations and remotely sensed aerosol optical depth. Concentrations are underestimated in Asia, but the higher emissions in CEDS than previous inventories result in improvements compared to observations. The treatment of black carbon (BC) scavenging in OsloCTM3 gives better agreement with observed vertical BC profiles relative to the predecessor OsloCTM2. However, Arctic wintertime BC concentrations remain underestimated, and a range of sensitivity tests indicate that better physical understanding of processes associated with atmospheric BC processing is required to simultaneously reproduce both the observed features. Uncertainties in model input data, resolution and scavenging affect the distribution of all aerosols species, especially at high latitudes and altitudes. However, we find no evidence of consistently better model performance across all observables and regions in the sensitivity tests than in the baseline configuration.

Using CEDS, we estimate a net RFari in 2014 relative to 1750 of -0.17 W m$^{-2}$, significantly weaker than the IPCC AR5 2011-1750 estimate. Differences are attributable to several factors, including stronger absorption by organic aerosol, updated parameterization of BC absorption, and reduced sulfate cooling. The trend towards a weaker RFari over recent years is more pronounced than in the IPCC AR5, illustrating the importance of capturing recent regional emission changes.

## 1 Introduction

Changes in anthropogenic emissions over the industrial period have significantly altered the abundance, composition and properties of atmospheric aerosols, causing a change in the radiative energy balance. The net energy balance change is determined by a complex interplay of different types of aerosols and their interactions with radiation and clouds, causing both positive (warming) and negative (cooling) radiative impacts. Global aerosols were estimated by the Intergovernmental Panel on Climate Change fifth assessment report (IPCC AR5) to have caused an effective radiative forcing (ERF) of -0.9 W m$^{-2}$ over the industrial era from 1750 to 2011, but with considerable uncertainty (-1.9 to -0.1 W m$^{-2}$) [*Boucher et al.*, 2013]. This large uncertainty range arises from a number of factors, including uncertainties in emissions and the simulated spatiotemporal distribution of aerosols, their chemical composition and properties.

Historical emission estimates for anthropogenic aerosol and precursor compounds are key data needed for climate and atmospheric chemistry transport models in order to examine how these drivers have contributed to climate change. The Community Emissions Data System (CEDS) recently published a new time series of emissions from 1750 to 2014, which will be used in the upcoming CMIP6 [*Hoesly et al.*, 2018]. CEDS includes several improvements, including annual temporal resolution with seasonal cycles, consistent methodology between different species, and extending the time series to more recent years, compared to previous inventories and assessments [e.g., *Lamarque et al.*, 2010; *Taylor et al.*, 2012]. During the period from 2000 to 2014, global emissions of black carbon (BC) and organic carbon (OC) have increased, while nitrogen oxide (NOx) emissions have been relatively constant after 2008, and sulfur dioxide (SO$_2$) emissions were back at 2000 levels in 2014, after a temporary increase [*Hoesly et al.*, 2018]. Furthermore, both CEDS and other recent emission inventories report considerably higher estimates of global BC and OC emissions in recent years than earlier inventories [*Granier et al.*, 2011; *Klimont et al.*, 2017; *Lamarque et al.*, 2010; *Wang et al.*, 2014]. The global trend in emissions is driven by a strong increase in emissions from Asia and Africa, and a decline in North America and Europe. Capturing such geographical differences is essential, as the distribution, lifetime and radiative forcing of aerosols depend on their location.

After emission or formation, particles undergo transport, mixing, chemical aging and removal by dry and wet deposition, resulting in a short atmospheric residence time, and a highly heterogeneous distribution in space and time. Consequently, accurate representation of observed aerosols remains challenging, and previous studies have shown that considerable diversity in the abundance and distribution of aerosols exist between global models. *Bian et al.* [2017] found that the atmospheric burden of nitrate aerosols differ by a factor of 13 between the models in AeroCom Phase III, caused by differences in both chemical and deposition processes. A smaller, but still considerable, model spread in the simulated burden of organic aerosols (OA) from 0.6-3.8 Tg was found by *Tsigaridis et al.* [2014]. It was also shown that OA concentrations on average were underestimated. There has been particular focus on BC aerosols over recent years. Multi-model studies have shown

variations in global BC burden and lifetime up to a factor of 4-5 [*Lee et al.*, 2013; *Samset et al.*, 2014]. Previous comparisons of modeled BC distributions with observations have also pointed to two distinct features common to many models: an overestimation of high altitude concentrations at low- to mid-latitudes and discrepancies in the magnitude and seasonal cycle of high-latitude surface concentrations (e.g., [*Eckhardt et al.*, 2015; *Lee et al.*, 2013; *Samset et al.*, 2014; *Schwarz et al.*, 2013]. As accurate representation of the observed aerosol distributions in global models is crucial for confidence in estimates of radiative forcing (RF), these issues emphasize the need for broad and up-to-date evaluation of model performance.

The diversity of simulated aerosol distributions, and discrepancies between models and measurements, stem from uncertainties in the model representation aerosol processing. Knowledge of the factors that control the atmospheric distributions is therefore needed to identify potential model improvements and need for further observational data, and to assess how remaining uncertainties affect the modeled aerosol abundances and, in turn, estimates of RF and climate impact. A number of recent studies have investigated the impact of changes in aging and scavenging processes on the BC distribution, focusing on aging and wet scavenging processes (e.g., [*Bourgeois and Bey*, 2011; *Browse et al.*, 2012; *Fan et al.*, 2012; *Hodnebrog et al.*, 2014; *Kipling et al.*, 2013; *Lund et al.*, 2017; *Mahmood et al.*, 2016]), resulting in notable improvements, at least for specific regions or observational data sets. However, with some notable exceptions [e.g., *Kipling et al.*, 2016], few studies have focused on impacts of scavenging and other processes on a broader set of aerosol species or the combined impact in terms of total aerosol optical depth (AOD).

Here we use the CEDS historical emission inventory as input to the chemistry-transport model OsloCTM3 to quantify the change in atmospheric concentrations over the period of 1750 to 2014. The OsloCTM3 is an update of the OsloCTM2, and includes several key changes compared to its predecessor. The significant existing model spread and sensitivity to process parameterizations underlines the need for careful and updated documentation of new model versions, and the increasing amount of available measurement data allows for improved evaluation. Before the model is used to quantify historical time series, we therefore evaluate the simulated present-day aerosol concentrations and optical depth against a range of observations. To get a first-order overview of how uncertainties in key processes and parameters affect the atmospheric abundance and distribution of aerosols in the OsloCTM3, we perform a range of sensitivity simulations. In addition to changes in the scavenging (solubility) assumptions, runs are performed with different emission inventories, horizontal resolution, and meteorological data. The impact on individual species and total AOD, as well as on the model performance compared with observations, is investigated. Finally, we present updated estimates of the historical evolution of radiative forcing due to aerosol-radiation interactions from pre-industrial to present, taking into account recent literature on aerosol optical properties. Section 2 describes the model and methods, while results are presented in Sect. 3 and discussed in Sect. 4. The conclusions are given in Sect. 5.

139

## 2 Methods

141

### 2.1 OsloCTM3

143

The OsloCTM3 is an offline global 3-dimensional chemistry-transport model driven by 3-hourly meteorological forecast data [*Søvde et al.*, 2012]. The OsloCTM3 has evolved from its predecessor OsloCTM2 and includes several updates to the convection, advection, photodissociation and scavenging schemes. Compared with OsloCTM2, the OsloCTM3 has a faster transport scheme, an improved wet scavenging scheme for large scale precipitation, updated photolysis rates and a new lightning parameterization. The main updates and subsequent effects on gas-phase chemistry were described in detail in *Søvde et al.* [2012]. Here we document the aerosols in OsloCTM3, including BC, primary and secondary organic aerosols (POA, SOA), sulfate, nitrate, dust and sea salt. The aerosol modules in OsloCTM3 are generally inherited and updated from OsloCTM2. The following paragraph briefly describes the parameterizations, including updates new to this work.

154

The carbonaceous aerosol module was first introduced by *Berntsen et al.* [2006] and has later been updated with snow deposition diagnostics [*Skeie et al.*, 2011]. The module is a bulk scheme, with aerosols characterized by total mass and aging represented by transfer from hydrophobic to hydrophilic mode at a constant rate. In the early model versions, this constant rate was given by a global exponential decay of 1.15 days. More recently, latitudinal and seasonal variation in transfer rates based on simulations with the microphysical aerosol parameterization M7 were included [*Lund and Berntsen*, 2012; *Skeie et al.*, 2011]. Previous to this study, additional M7 simulations have been performed to include a finer spatial and temporal resolution in these transfer rates. Specifically, the latitudinal transfer rates have been established based on experiments with 10 instead of four emission source regions and with monthly, not seasonal resolution. In OsloCTM3 the carbonaceous aerosols from fossil fuel and biofuel combustion are treated separately, allowing us to capture differences in optical properties in subsequent radiative transfer calculations (Sect. 2.4). In contrast to the OsloCTM2, OsloCTM3 treats POA instead of OC. If emissions are given as OC, a factor of 1.6 for anthropogenic emissions and 2.6 for biomass burning sources is used for the OC-to-POA conversion, following suggestions from observational studies [*Aiken et al.*, 2008; *Turpin and Lim*, 2001]. Upon emission, 20% of BC is assumed to be hydrophilic and 80% hydrophobic, while a 50/50 split is assumed for POA [*Cooke et al.*, 1999]. An additional update in this work is the inclusion of marine primary organic aerosols following the methodology by *Gantt et al.* [2015], where emissions are determined by production of sea spray aerosols and oceanic chlorophyll A. Monthly concentrations of chlorophyll A from the same year as the meteorological data is taken from the Moderate Resolution Imaging Spectroradiometer (MODIS; available from https://modis.gsfc.nasa.gov/data/dataprod/chlor_a.php ), while sea spray aerosols are simulated by the OsloCTM3 sea salt module. The climatological annual mean total emission

of marine POA is scaled to 6.3 Tg based on *Gantt et al.* [2015]. The scaling factor depends on the
chosen sea salt production scheme (described below) and to some degree on the resolution; here
we have used a factor of 0.5.
The formation, transport and deposition of SOA are parameterized as described by *Hoyle et al.*
[2007]. A two product model (Hoffmann et al., 1997) is used to represent the oxidation products
of the precursor hydrocarbons and their aerosol forming properties. Precursor hydrocarbons which
are oxidized to form condensable species include both biogenic species such as terpenes and
isoprene, as well as species emitted predominantly by anthropogenic activities (toluene, m-xylene,
methylbenzene and other aromatics). The gas/aerosol partitioning of semi-volatile inorganic
aerosols is treated with a thermodynamic model [*Myhre et al.*, 2006]. The chemical equilibrium
between inorganic species (ammonium, sodium, sulfate, nitrate and chlorine) is simulated with the
Equilibrium Simplified Aerosol model (EQSAM) [*Metzger et al.*, 2002a; *Metzger et al.*, 2002b].
The aerosols are assumed to be metastable, internally mixed and obey thermodynamic gas/aerosol
equilibrium. Nitrate and ammonium aerosols are represented by a fine mode, associated with sulfur,
and a coarse mode associated with sea salt, and it is assumed that sulfate and sea salt do not interact
through chemical equilibrium [*Myhre et al.*, 2006]. The sulfur cycle chemistry scheme and
aqueous-phase oxidation is described by *Berglen et al.* [2004].
The sea salt module originally introduced by *Grini et al.* [2002] has been updated with a new
production parameterization following recommendations by *Witek et al.* [2016]. Using satellite
retrievals, Witek et al. (2016) evaluated different sea spray aerosol emission parametrizations and
found the best agreement with the emission function from *Sofiev et al.* [2011] including the sea
surface temperature adjustment from *Jaeglé et al.* [2011]. Compared to the previous scheme, the
global production of sea salt is reduced, while there is an increase in the tropics. This will have an
impact on the uptake of nitric acid in sea salt particles, consequently affecting NOx, hydroxide
(OH) and ozone levels. However, here we limit the scope to aerosols. The Dust Entrainment and
Deposition (DEAD) model v1.3 [*Zender et al.*, 2003] was implemented into OsloCTM2 by *Grini*
*et al.* [2005] and is also used in OsloCTM3. As a minor update, radiative flux calculations, required
for determination of boundary layer properties in the dust mobilization parameterization [*Zender*
*et al.*, 2003], now uses radiative surface properties and soil moisture from the meteorological fields.
Aerosol removal includes dry deposition and washout by convective and large-scale rain. Rainfall
is calculated based on European Center for Medium-Range Weather Forecast (ECMWF) data for
convective activity, cloud fraction and rain fall. The efficiency with which aerosols are scavenged
by the precipitation in a grid box is determined by a fixed fraction representing the fraction of this
box that is available for removal, while the rest is assumed to be hydrophobic. The
parameterization distinguishes between large-scale precipitation in the ice and liquid phase, and
the OsloCTM3 has a more complex cloud model than OsloCTM2 that accounts for overlapping
clouds and rain based on *Neu and Prather* [2012]. When rain containing species falls into a grid
box with drier air it will experience reversible evaporation. Ice scavenging, on the other hand, can
be either reversible or irreversible. For further details about large-scale removal we refer the reader
to *Neu and Prather* [2012]. Convective scavenging is based on the Tiedtke mass flux scheme
(Tiedtke 1989) and is unchanged from the OsloCTM2. The solubility of aerosols is given by
constant fractions, given for each species and type of precipitation (i.e., large-scale rain, large-
scale ice, and convective) (Table 2). Dry deposition rates are unchanged from OsloCTM2, but the
OsloCTM3 includes a more detailed land use dataset (18 land surface categories at 1°x1°
horizontal resolution compared to 5 categories at T42 resolution), which affects the weighting of
deposition rates for different vegetation categories. Re-suspension of dry deposited aerosols is not
treated.
2.2 Emissions

The baseline and historical simulations use the CEDS anthropogenic [*Hoesly et al.*, 2018; *Smith et
al.*, 2015] and biomass burning (BB4CMIP) [*van Marle et al.*, 2017] emissions. The CEDS
inventory provide monthly gridded emissions of climate-relevant greenhouse gases, aerosols and
precursor species from 1750 to 2014 using a consistent methodology over time. Anthropogenic
CEDS emissions are comparable to, but generally higher than, other existing inventories [*Hoesly
et al.*, 2018]. Biogenic emissions are from the inventory developed with the Model of Emissions
of Gases and Aerosols from Nature under the Monitoring Atmospheric Composition and Climate
project (MEGAN-MACC) [*Sindelarova et al.*, 2014] and are held constant at the year 2010 level.
Here we use the CEDS version released in 2016 (hereafter CEDSv16). In May 2017, after
completion of our historical simulations, an updated version of the CEDS emission inventory was
released after users reported year-to-year inconsistencies in the country/sector level gridded data.
The emission totals were not affected, but there were occasional shifts in the distribution within
countries (http://www.globalchange.umd.edu/ceds/ceds-cmip6-data/). The potential implications
for our simulations are discussed below.
Two other emission inventories are also used. The ECLIPSEv5 emission dataset was created with
the Greenhouse Gas - Air Pollution Interactions and Synergies (GAINS) model [*Amann et al.*,
2011] and provides emissions in 5 year intervals from 1990 to 2015, as well as projections to 2050
[*Klimont et al.*, 2017]. The 1990-2015 emission series was recently used to simulate changes in
aerosols and ozone and their RF [*Myhre et al.*, 2017]. Here we only use emissions for 2010 in the
sensitivity simulation.
The Representative Concentration Pathways (RCPs) [*van Vuuren et al.*, 2011] were developed as
a basis for near- and long-term climate modeling and were used in CMIP5 and Atmospheric
Chemistry and Climate Model Intercomparison Project (ACCMIP) experiments. While the four
RCPs span a large range in year 2100 RF, emissions of most species have not diverged
significantly in 2010 and we select the RCP4.5 for use here [*Thomson et al.*, 2011]. Table S1
summarized total global emissions of BC, OC, $NO_x$ and $SO_2$ in 2010 in each of the three scenarios.
In the simulations with the ECLIPSEv5 and RCP4.5 inventories, biomass burning emissions are
from the Global Fire Emission Database Version 4 (GFED4) [*Randerson et al.*, 2017]. The
BB4CMIP emissions are constructed with GFED4 1997-2015 emissions as a basis [*van Marle et*
*al.*, 2017] and emissions in 2010 are similar in both datasets. Hence, any difference between the
sensitivity simulations stems from differences in the anthropogenic inventory.

2.3 Simulations

Time slice simulations with CEDSv16 emissions for 1750, 1850 and from 1900 to 2014 are
performed (every ten years from 1900-1980, thereafter every five years), one year with six months
spin-up. The model is run with fixed year 2010 meteorological data and a horizontal resolution of
2.25x2.25 degrees (denoted 2x2), with 60 vertical layers. While *Søvde et al.* [2012] used
meteorological data from the ECMWF IFS model cycle 36r1, we apply here meteorology from the
ECMWF OpenIFS cycle 38r1 (https://software.ecmwf.int/wiki/display/OIFS/).

Additional model runs are performed to investigate the importance of differences in key processes
for the aerosol distributions and model performance (Table 1). In addition to the CEDSv16
emissions, the model is run with ECLIPSEv5 and RCP4.5 emission inventories for anthropogenic
emissions and GFEDv4 biomass burning emissions. Additionally, we perform simulations with
1.125x1.125 degrees (denoted 1x1) horizontal resolution. To investigate the importance of
meteorology, the simulation with CEDSv16 emissions is repeated with meteorological data for
year 2000 instead of 2010. Year 2000 is selected due to its opposite El Niño–Southern Oscillation
(ENSO) index compared to 2010. Finally, three model runs are performed with increased and
decreased aerosol removal by large-scale ice clouds and decreased aerosol scavenging by liquid
(large-scale and convective) precipitation. To modify the scavenging, we tune the fixed fractions
that control aerosol removal efficiency in the model (see Sect. 2.1). Table 2 summarizes fractions
used in the baseline configuration and the three sensitivity tests. A decrease and increase in
efficiency of 0.2 is adopted for scavenging of all aerosols by liquid clouds (except hydrophobic
BC and POA) and ice clouds, respectively. Note that there is no test with increased removal by
liquid clouds, as, with the exception of hydrophobic BC, POA and SOA, 100% efficiency is
already assumed. For ice clouds we also reduce the efficiency to a fraction of 0.1, or 0.001 if the
value is 0.1 in the baseline configuration. We note that these changes do not represent realistic
uncertainty ranges based on experimental or observational evidence, as there are limited
constraints in the literature, but are chosen to explore the impact of a spread in the efficiency with
which aerosols act as ice and cloud condensation nuclei.

2.4 Radiative transfer

We calculate the instantaneous top-of-the atmosphere radiative forcing of anthropogenic aerosols
due to aerosol-radiation interactions (RFari) [*Myhre et al.*, 2013b]). The radiative transfer

calculations are performed offline with a multi-stream model using the discrete ordinate method [*Stamnes et al.*, 1988]. The model includes gas absorption, Rayleigh scattering, absorption and scattering by aerosols, and scattering by clouds. The RFari of individual aerosols is obtained by separate simulations, where the concentration of the respective species is set to the pre-industrial level. The aerosol optical properties have been updated from earlier calculations using this radiative transfer model [*Myhre et al.*, 2007; *Myhre et al.*, 2009], in particular those associated with aerosol absorption. The *Bond and Bergstrom* [2006] recommendation of a mass absorption coefficient (MAC) for BC of around 7.5 $m^2$ $g^{-1}$ for freshly emitted BC and an enhancement factor of 1.5 for aged BC was used previously. In the present analysis, we apply a parametrization of MAC from observations over Europe by *Zanatta et al.* [2016], where MAC depends on the ratio of non-BC to BC abundance. The mean MAC of BC from these observations around 10 $m^2$ $g^{-1}$ at 630 nm [*Zanatta et al.*, 2016]. The measurements in *Zanatta et al.* [2016] represent continental European levels. For very low concentrations of BC, the formula given in *Zanatta et al.* [2016] provides very high MAC values. We have therefore set a minimum level of BC of 1.0e-10 g $m^{-3}$ for using this parameterization, and for lower concentrations we use *Bond and Bergstrom* [2006]. In addition, we have set a maximum value of MAC of 15 $m^2$ $g^{-1}$ (637 nm) to avoid unrealistic high values of MAC compared to observed values. Organic matter has a large variation in the degree of absorption [e.g., Kirchstetter et al., 2004; Xie et al., 2017], from almost no absorption to a strong absorption in the ultraviolet region. Here, we have implemented absorbing organic matter according to refractive indices from Kirchstetter et al. [2004]. The degree of absorption varies by source and region and is at present inadequate quantified: Here we assume 1/3 of the biofuel organic matter and ½ of the SOA from anthropogenic volatile organic carbon (VOC) precursors. The remaining fractions of biofuel, fossil fuel and marine POA and SOA (anthropogenic and all natural VOCs) are assumed to be purely scattering organic matter. As these fractions are not sufficiently constrained by observational data and associated with significant uncertainty, we also perform calculations with no absorption by organic matter for comparison.

2.5 Observations

A range of observational datasets are used to evaluate the model performance in the baseline simulation. Note that we use the term "black carbon" in a qualitative manner throughout the manuscript to refer to light-absorbing carbonaceous aerosols. However, when comparing with measurements, we use either elemental carbon (EC) or refractive BC (rBC), depending on whether the data is derived from methods specific to the carbon content of carbonaceous aerosols or incandescence methods, in line with recommendations from *Petzold et al.* [2013].

Measured surface concentrations of EC, OC, sulfate and nitrate are obtained from various networks. For the US, measurements from IMPROVE (Interagency Monitoring of Protected Visual Environments) and CASTNET (Clean Air Status and Trends Network) are used. For Europe, data from EMEP (European Monitoring and Evaluation Programme) [*Tørseth et al.*, 2012] and

ACTRIS (Aerosols, Clouds and Trace gases Research InfraStructure) [*Cavalli et al.*, 2016; *Putaud et al.*, 2010] is used. EMEP and ACTRIS sites are all regional background sites, representative for a larger area. To broaden the geographical coverage we also compare the model output against additional observations from the CMA Atmospheric Watch Network (CAWNET) in China [*Zhang et al.*, 2012] and those reported in the literature from India (see *Kumar et al.* [2015] for more details). CASTNET, IMPROVE, EMEP and ACTRIS data is from year 2010, while CAWNET observations were sampled in 2006-2007 and the observational data base from India compiled by *Kumar et al.* [2015] cover a range of years. IMPROVE provides mass of aerosols using filter analysis of measurements of particulate matter with diameter of less than 2.5 micrometers ($PM_{2.5}$), while CASTNET uses an open-face filter pack system with no size restriction to measure concentrations of atmospheric sulfur and nitrogen species [*Lavery et al.*, 2009]. Mass of individual species from the CAWNET network is obtained from aerosol chemical composition analysis performed on $PM_{10}$ samples [*Zhang et al.*, 2012]. EMEP and ACTRIS measurements of EC and OC are in the $PM_{2.5}$ range, whereas nitrate and sulfate measurements are filter-based with no size cutoff limit. Data resulting from EMEP and ACTRIS are archived in the EBAS data base (http://ebas.nilu.no) at NILU - Norwegian Institute for Air Research, and are openly available (see also Data availability).

Modeled AOD is evaluated against the Aerosol Robotics Network (AERONET). AERONET is a global network of stations measuring radiance at a range of wavelengths with ground-based sun-photometers, from which aerosol column burden and optical properties can be retrieved [*Dubovik and King*, 2000; *Holben et al.*, 1998]. The comparison with AERONET data was done using the validation tools available from the AeroCom data base hosted by Met Norway (http://aerocom.met.no/data.html). We also compare against AOD retrievals from MODIS-Aqua and Terra (level 3 atmosphere products, AOD550 combined dark target and deep blue, product version 6) [*MOD08*, 2018] and the Multi-angle Imaging SpectroRadiometer (MISR) (level 2 aerosol product, product version 4) [*MISR*, 2018].

Figure S1 depicts the locations of all the stations. For comparison with surface concentrations and AERONET AOD, the model data is linearly interpolated to the location of each station using annual mean, monthly mean (concentrations) or 3-hourly output (AOD), depending on the resolution of the observations. In the case of AERONET, high mountain stations (here defined as having an elevation higher than 1000 meter above sea level) are excluded following *Kinne et al.* [2013]. For comparison with observed OC surface concentrations, modeled OA is converted to OC using factor of 1.6 for POA and 1.8 for SOA. Unless measurements are restricted to the PM2.5 size range, the comparison includes both fine and coarse mode modeled nitrate (Sect. 2.1). Several statistical metrics are used to assess the model skill, including correlation coefficient (R), root mean square error (RMSE), variance and normalized mean bias (NMB).

The modeled vertical distribution of BC is compared with aircraft measurements of refractory BC (rBC) from the HIAPER Pole-to-Pole Observations (HIPPO) campaign [*Wofsy et al.*, 2011].

Vertical profiles of BC from OsloCTM2 have been evaluated in several previous studies (e.g.,
*Samset et al.* [2014]) and a more thorough comparison of OsloCTM3 results against a broader set
of campaigns is provided by *Lund et al.* [2018]. In the present analysis we focus on data from the
third phase (HIPPO3) flights, the only phase that was conducted in 2010, i.e., the same year as our
sensitivity simulations. Model data is extracted along the flight track using an online flight
simulator. The data is separated into five latitude regions and vertical profiles constructed by
averaging observations and model output in 13 altitude bins.

3 Results

We first document the aerosol distributions simulated in the baseline model configuration,
focusing on the anthropogenic contribution, and compare with observations, multi-model studies
and results from the sensitivity tests. With the present-day model performance evaluated, we then
present the updated historical development of RFari of anthropogenic aerosols.

3.1 Evaluation of present-day aerosol distributions

The global mean aerosol burdens and atmospheric residence times (ratio of burden to total wet
plus dry deposition) in the baseline simulation are summarized in Table 3 (top row), with spatial
distribution shown in Fig. S2. Compared to results from the AeroCom III experiment, the
OsloCTM3 sulfate burden of 5.4 mg m$^{-2}$ estimated here is about 50% higher than the multi-model
mean of 3.5 mg m$^{-2}$ and 35% higher than OsloCTM2 [*Bian et al.*, 2017]. While the total SO$_2$
emission is only 5% higher in the present study than in the OsloCTM2 AeroCom III simulation,
the atmospheric residence time of sulfate is 50% longer, suggesting that the burden difference is
mainly attributable to changes in the parameterization of dry and large-scale wet deposition in
OsloCTM3 (Sect. 2.1). The nitrate burden is nearly a factor three higher than both the AeroCom
multi-model mean and OsloCTM2 burden, and higher than all nine models contributing in
AeroCom III [*Bian et al.*, 2017]. This is mainly due to a higher burden of coarse mode nitrate
aerosols, associated with less efficient scavenging of sea salt in OsloCTM3 than OsloCTM2. The
global budgets of OA simulated by the AeroCom II models was analyzed by *Tsigaridis et al.*
[2014]. The burden of OA in the OsloCTM3 of 3.4 mg m$^{-2}$ is close to their multi-model mean of
3.1 mg m$^{-2}$ and 25% higher than the OsloCTM2. The OsloCTM3 estimate includes the contribution
from marine OA emissions (Sect. 2.1), which may explain part of the difference as marine OA was
included in some of the AeroCom II models, but not OsloCTM2. However, the marine POA only
contributes around 3% to the total OA. Additionally, the residence time of OA of 5.3 days is longer
than in the OsloCTM2 AeroCom II experiment. The global BC burden of 0.23 mg m$^{-2}$ is also close
to the mean of the AeroCom II models of 0.25 mg m$^{-2}$ [*Samset et al.*, 2014]. We note that different
emission inventories were used in the AeroCom experiments and the present analysis, however,

the comparison shows that the aerosol burdens simulated by OsloCTM3 fall within the range of existing estimates from global models.

Figure 1 shows results from the baseline OsloCTM3 simulation against annual mean measured surface concentrations of EC, OC, sulfate and nitrate in Europe, North America and Asia. Overall, the OsloCTM3 shows a high correlation of 0.8-0.9 with measured surface concentrations. There is a general tendency of underestimation by the model, with the lowest NMB and RMSE for BC and nitrate (-23%) and the highest for sulfate (-51%). There are, however, notable differences in model performance between data sets in different regions, as seen from Table S2. For all species, the NMB and RMSE are highest for measurements in China. For instance, excluding the CAWNET measurements, reduces the NMB for sulfate in Fig. 1 from -51% to -31% (not shown). In contrast, the correlation with CAWNET observations is generally similar to, or higher than, other regions/networks. In the case of BC and nitrate, the model slightly overestimates concentrations in Europe and North America, but underestimates Asian measurements. The best overall agreement is generally with IMPROVE observations in North America. Differences in instrumentation between different networks can affect the evaluation. *Lavery et al.* [2009] found that measurements from CASTNET typically gave higher nitrate surface concentrations than values obtained from co-located IMPROVE stations, which could partly explain the NMB of opposite sign in these two networks in Table S2. For BC, we also include measurements from across India compiled by *Kumar et al.* [2015]. This is a region where emissions have increased strongly, but where evaluation of the model performance so far has been limited due to availability of observations. The model underestimates concentrations with a NMB of -43%, however, the correlation of 0.60 is similar to the comparison with data from China and higher than the other regions. An examination of the monthly concentrations (Fig. S3) shows that the largest discrepancies occur during winter, with the largest bias found for measurements in North East India. One possible reason could be missing or underestimated emission sources. This finding is similar to the comparison of measurements against WRF-chem by *Kumar et al.* [2015]. The seasonality of BC concentrations has also been an issue at high northern latitudes, where earlier versions of the OsloCTM strongly underestimated winter and springtime surface concentrations at Arctic stations [*Lund et al.*, 2017; *Skeie et al.*, 2011], similar to many other models [*Eckhardt et al.*, 2015]. This Arctic underestimation persists in the current version of the model. Seasonal differences exist also in other regions, but not consistently across measurement networks. Compared with EC measurements from EMEP/ACTRIS the correlation is poorer during winter and spring, and the model underestimate concentrations in contrast to a positive NMB in summer and fall. However, due to the relatively low number of stations, these values are sensitive to a few stations with larger measurement-model discrepancies. For both IMPROVE and EMEP/ACTRIS, the model underestimation of sulfate is larger during summer and fall, but with opposite seasonal differences in correlation. In general, the number of stations and evaluation of data from only one year limits the analysis of seasonal variations.

We do not evaluate ammonium concentrations in the present analysis, as that requires a detailed
discussion of the nitrate and sulfate budgets, which has been covered by the recent multi-model
evaluation by *Bian et al.* [2017] based on an AeroCom Phase III experiment, in which the
OsloCTM3 participated. Results showed that most models tend to underestimate ammonium
concentrations compared to observations in North America, Europe and East Asia, with a multi-
model mean bias and correlation of 0.886 and 0.47, respectively. The OsloCTM3 shows good
agreement with ammonium measurements in North America, but has a bias and correlation close
the model average in the other two regions.
In May 2017, after completion of our historical simulations, an updated version of the CEDS
emission inventory was released after an error in the code was reported (see Sect. 2.2). This
resulted in occasional shifts in the spatial distribution of emissions within countries with large
spatial extent (e.g., USA and China). Since the emission totals were not affected, the impact on
our RFari estimates is likely to be small, but shifts in the emission distribution could influence the
model evaluation, in particular for surface concentrations. While repeating all simulations would
require more resources, we have repeated the year 2010 and 1750 runs. Figure S4 shows the
comparison of modeled concentrations against IMPROVE measurements with the two emission
inventory versions, CEDSv16 and CEDSv17. In the case of BC, the comparison shows a 5% higher
correlation and 15% lower RMSE with the CEDSv17 than CEDSv16. A similar improvement is
found for nitrate, with 26% higher correlation and 12% lower RMSE, while in the case of OC and
sulfate, the difference is small ($< 5\%$). Smaller differences of between 2-10% are also found in the
comparison against measurements in Europe and Asia (not shown). Hence, using the updated
version of the emission inventory has an effect on the model performance in terms of surface
concentrations, but without changing the overall features or conclusions. The net RFari in 2010
relative to 1750 is 2% stronger with the CEDSv17 inventory, a combined effect of slightly higher
global BC burden and lower burdens of sulfate and OA.
As shown in Table S2, the model overestimate surface concentrations in some regions and
underestimate them in others. Compensating biases can influence the evaluation of total AOD.
Moreover, the biases differ in magnitude between different species. Moving one step further, we
therefore examine the average aerosol composition in the three regions where this is possible with
our available measurements. Figure 2 shows the relative contribution from different aerosols
species to the total mass in the IMPROVE, EMEP, ACTRIS and CAWNET measurements and the
corresponding model results. The number of available aerosol species varies between the
measurement networks and we include sea salt from IMPROVE and ammonium from CAWNET.
Additionally, the number of stations where simultaneous measurements of all species were
available also differ substantially, with 16 for CAWNET, 5 for EMEP/ACTRIS and 172 for
IMPROVE. Overall, the relative composition is well represented by the model. The agreement is
particularly good for the IMPROVE network. Compared to measurements from CAWNET, the
model has a lower relative contribution from OC and more sulfate. In the case of Europe, nitrate
aerosols also constitute a significantly larger fraction in the model than in the observations. The
evaluation of nitrate is complicated by possible differences in the detection range of
instrumentation compared to the size of the two nitrate modes in the model (Sect. 2.1). The
comparison against EMEP nitrate data includes both coarse and fine mode modeled nitrate.
Excluding the coarse mode, the fraction of total mass attributable to nitrate decreases from 43% to
28%, which is much closer to the observed 30% contribution. However, this affects the comparison
in Figure 1, resulting in a negative NMB of -34%, compared to -23% when including both coarse
and fine mode. This suggest that part, but not all, of the nitrate represented as a coarse mode in the
model is measured by the instrument, pointing to a need for a more sophisticated size distribution
in the model to make better use of available observations. The low number of available stations
from EMEP/ACTRIS could also an important factor.
Next, we examine total AOD. Figure 3 shows modeled AOD and aerosol absorption optical depth
(AAOD), AOD retrieved from MODIS-Aqua and comparison of modeled AOD with AERONET
observations. Modeled global, annual mean AOD and AAOD is 0.13 (Fig. 3a) and 0.005 (Fig. 3b),
respectively. The overall spatial pattern of modeled AOD agrees well with MODIS (Fig. 3c),
however, the latter gives a higher global mean of 0.16 and clearly higher values in North India and
parts of China, as well as Central Africa. These peak values are similar to MODIS-Terra, but less
pronounced in the AOD retrieved from MISR (Fig. S5), illustrating important differences between
different remote sensing products. Nevertheless, an underestimation of modeled AOD in Asia is
consistent with results from the evaluation of surface concentrations and can also be seen in the
comparison against AERONET, as discussed below. The OsloCTM3 shows a good agreement
with measured AOD from the AERONET network, with an overall correlation of 0.82 and RMSE
of 0.11, when using monthly mean data from 266 stations (Fig. 3d). Note that the modeled global
mean AOD is 0.13, but the model mean at the AERONET stations is 0.175 (Fig 3d) and has only
a NMB of -11.8%. Many of the AERONET stations tend not to be regional background sites, but
can be influenced by local pollution (e.g., *Wang et al.* [2018])
There are notable regional differences in model performance. Fig. S6 compares modeled AOD
against AERONET stations in Europe, North America, India and China separately. The best
agreement is found for Europe and North America, with NMB of -0.4% and -13%, respectively,
and RMSE of approx. 0.05. The correlation is higher for North America (0.76) than Europe (0.63).
A relatively high correlation of 0.71 is also found for stations in China. However, the NMB and
RMSE is higher (-34.5% and 0.25). There are significantly fewer observations for comparison with
modeled AOD over India, but the ones available give NMB and RMSE on the same order of
magnitude as for China, but a lower correlation (0.45).
Ground-based measurements can also provide information about column absorption aerosol
optical depth (AAOD). Such information has been used to constrain the absorption of BC and
provide top-down estimate of the direct BC RF (e.g., [*Bond et al.*, 2013]). However, retrieval and
application of AERONET AAOD is associated with a number of challenges and uncertainties (e.g.,
[*Samset et al.*, 2018]), hence such an evaluation is not performed here.
Recent literature has pointed to important representativeness errors arising when constraining
models using observations due to the coarse spatial and temporal scales of global models compared
with the heterogeneity of observations. *Schutgens et al.* [2016a] found differences in RMSE of up
to 100% for aerosol optical thickness when aggregating high resolution model output over grid
boxes representative of the resolution of current global models compared to small areas
corresponding to satellite pixels. Smaller, but notable, differences of up to 20% were found when
monthly mean model data was used, as in the present analysis. However, that did not account for
issues related to temporal collocation, which can also introduce considerable errors [*Schutgens et
al.*, 2016b]. In a recent study, *Wang et al.* [2018] found a spatial representativeness error of 30%
when constraining AAOD modeled at a 2°x2° horizontal resolution against AERONET retrievals.
However, further work is needed to investigate whether similar biases exist for AOD.

3.2 Sensitivity of aerosols distributions to model input and process parameterization

As shown in the section above, the OsloCTM3 performs well compared against observed AOD.
Still, a number of factors influence the simulated distributions of individual aerosol species. To
assess the importance of key uncertainties for modeled distributions and model performance, we
perform a range of sensitivity simulations (Table 1) to examine the importance of emission
inventory, scavenging assumptions (Table 2), meteorological data and resolution for the modeled
aerosol distributions and model performance.

Global aerosol burdens and AOD in each sensitivity run are summarized in Table 3 (corresponding
atmospheric residence times are given in Table S3). The BC burden is particularly sensitive to
reduced scavenging by large-scale ice clouds (LSIDEC), resulting in a 40% higher burden
compared to the baseline. In contrast, an equal increase in the scavenging efficiency (LSIINC)
result in a decrease in burden of only 9%, while decreased scavenging by liquid precipitation
(SOLDEC) gives a 13% higher burden. The lower BC emissions in the ECLv5 and CMIP5
inventories give a global BC burden that is 9 and 22% lower, respectively. For sulfate, ammonium
and OA, we also find the largest burden changes in the LSIDEC case, followed by SOLDEC. The
change in the LSIDEC is particularly large for OA and is driven by changes in SOA. For SOA, the
changes are determined not only by modifying the scavenging, but also by changes in POA
concentrations, which gas-phase secondary organics can partition onto. Increasing the horizontal
resolution results in a slightly higher burden for all species, except sea salt.
While sensitivity tests may give similar changes in the total global burdens, the spatial distribution
of changes can differ substantially. Figure 4 shows the ratio of AOD and total burden by species
and altitude in each sensitivity simulation to the baseline. As expected, varying the emission
inventories results in changes that are largely confined to the main source regions (Figs.4a,b).
Using the CMIP5 inventory results in considerably lower concentrations over Asia, the Middle
East and North Africa, reflecting the higher emissions in the more recent inventory. Over central
North America the AOD is higher, mainly due to more ammonium nitrate, whereas the higher
AOD over Eastern Europe and part of Russia is a result of higher sulfate concentrations. Similar
characteristics are found when using ECLv5, but the relative differences are smaller. Reducing or
increasing the large-scale ice cloud scavenging gives the largest relative changes in AOD at high
latitudes, while changes in the solubility assumption for liquid precipitation affects AOD most
over Asia, where aerosol burdens are high, and around the equator where convective activity is
strong. In general, the burden of BC, OA and dust is significantly affected by changes in the
scavenging assumptions, while nitrate responds more strongly to different emission inventories,
likely due to the complicated dependence on emissions of several precursors and competition with
ammonium-sulfate. We also note that at higher altitudes the absolute differences in the burden of
nitrate are small. Changes in AOD resulting from using different meteorological input data are
more heterogeneous and are most notable in regions where effects of choosing data from years
with opposite ENSO phase are expected, e.g., west coast of South America and South East Asia.
There is also a notable change in the Atlantic Ocean, where mineral dust is a dominating species.
The meteorological data can affect production, deposition and transport of dust directly, as well as
indirectly through ENSO-induced teleconnections as suggested by e.g., *Parhi et al.* [2016].
For BC, OA and dust, the largest impact relative to the baseline are seen above 600 hPa in the
LSIDEC case. Change in LSIDEC are also important in the case of sulfate and sea salt, but occur
at lower altitudes. In contrast to the other aerosol species, differences in emission inventories are
most important for nitrate. In a recent study, *Kipling et al.* [2016] investigated factors controlling
the vertical distribution of aerosols in the HadGEM3-UKCA. It was found that in-cloud
scavenging was very important in controlling the vertical mass concentration of all species, except
dust. For dust, it was also found that dry deposition and sub-cloud processes played key roles,
processes not examined in the present analysis. Moreover, *Kipling et al.* [2016] performed
sensitivity simulations by switching transport and scavenging on and off to get the full effect of a
given process, while we perform smaller perturbations to investigate uncertainties. Here we find
significant impacts of changes in ice-cloud removal efficiency (Table 2) on the vertical distribution
of BC, OA and dust, while large-scale liquid and convective precipitation is more important for
sea salt and nitrate
Our sensitivity tests show that changes in input data, resolution or scavenging can lead to notable
changes in the aerosol distributions. The next question is then how these changes affect model
performance compared to observations. Figure 5a compares modeled and measured surface
concentrations of BC, OC, sulfate and nitrate in each simulation using all observations in Fig. 1.
For BC, the sensitivity tests have little or no impact on correlation, but there is a markedly better
agreement in terms of standard deviation (i.e., model becomes closer to observations) for
CEDSv16/CMIP6 compared to RCP/CMIP5, reflecting the higher emissions in the former. Similar,
but smaller, effects are also found for the other species. The improvement from RCP/CMIP5 to
CEDSv16/CMIP6 is especially seen for measurements in Asia. A higher resolution is also found
to reduce the bias, in particular for BC. Figure 5b shows the comparison against AERONET AOD
in each sensitivity simulation. Again, there is a higher correlation and lower bias in the 1x1RES
run than in the baseline, while the opposite is found in the RCP/CMIP5 and ECLv5 cases. For both
observables, the improvement in the 1x1RES simulation may result from a better sampling at a
finer resolution, improved spatial distribution or a combination. The most pronounced changes
results from using meteorological data from year 2000, in which case the correlation is reduced
from around 0.8 to 0.7.
For both observables, the difference in model performance between the baseline and scavenging
sensitivity tests is small. This may partly be an effect of the geographical coverage of stations; the
majority of measurements are from stations in more urban regions, whereas simulated burden
changes occur in remote regions, particularly at high latitudes and altitudes (Fig. 4). We therefore
also perform evaluations against AOD from regional sub-sets of AERONET stations. Ten of the
AERONET stations used in the present analysis are located north of 65°N (Fig. S1). A comparison
of monthly mean simulated AOD in each of the sensitivity runs against observations in this region
shows the best agreement with the baseline simulation and with the ECLv5 emission inventory,
with a considerably higher bias when scavenging parameters are modified (Fig. S7a). This is
particularly the case in the LSIDEC run, where concentrations of all species increase at high
latitudes compared to the baseline (Fig. 4). In contrast, the reduced concentrations in LSIINC,
results in a negative bias. We note that most of these stations have missing values in the winter
months, which is when the model underestimate BC concentrations in the Arctic, hence limiting
the evaluation. Decreased scavenging efficiency also leads to a higher bias than in the baseline for
observations in Europe and North America (not shown). In Asia, where the model already
underestimates aerosols in the baseline configuration, the bias is reduced since concentrations
increase. However, differences are smaller than north of 65°N. Moreover, given the notable
exacerbation in model performance in other regions, it is likely that other sources of uncertainty
(e.g., emissions) are more important for the model-measurement discrepancies in Asia. A similar
comparison is performed for 15 AERONET stations located in North Africa and the Middle East
(Fig. S7b), where the dust influence is strong. Changing the meteorological year and reducing
scavenging results in higher dust burdens (Table 3). Again, the agreement is better in the baseline
run than in these sensitivity runs. In particular, the METDATA run result in a higher bias and a
lower correlation, which is not surprising as dust production depends also on meteorological
conditions. The changes compared to the baseline CEDSv16/CMIP6 simulation cannot be entirely
attributed to differences in dust concentrations, as seen from the RCP/CMIP5 and ECLv5
simulations where the dust production is equal to the baseline. Several studies have pointed to the
importance of spatial resolution for improved model performance compared to observations (e.g.,
[*Sato et al.*, 2016; *Schutgens et al.*, 2017; *Schutgens et al.*, 2016a; *Wang et al.*, 2016]). *Wang et al.*
[2016] found significant reductions in NMB of BC AAOD relative to AERONET when using a
high resolution (10 km) emission data and model output. In our analysis, moving from 2°x2° to
1°x1° horizontal resolution also results in a slightly higher correlation and reduced bias and errors
when compared to all AERONET stations (Fig. 5b). The impact is largest for AOD in China and
India, the NMB is reduced (from -34% and -24% (Fig. S6) to -20% and -10%, respectively).
However, the opposite effect is found for AERONET stations in Europe and North America. Of
course, the 1°x1° resolution is still very coarse compared to the grid sizes used in the
abovementioned studies.
Changes away from near-source areas are also evaluated in terms of BC concentrations by a
comparison with observed vertical distribution from the HIPPO3 campaign, where remote, marine
air over the Pacific was sampled across all latitudes (Sect. 2.5). To limit the number of model runs,
we focus on only one phase of the HIPPO campaign here, but a more comprehensive evaluation
of OsloCTM3 vertical BC distribution against aircraft measurements was performed by *Lund et*
*al.* [2018]. Figure 6 shows observed average vertical BC concentration profiles against model
results from each sensitivity test. The OsloCTM3 reproduces the vertical distribution well in low
and mid-latitudes over the Pacific in its baseline configuration, although near-surface
concentrations in the tropics are underestimated. This is a significant improvement over the
OsloCTM2, where high-altitude concentrations in these regions typically were overestimated. The
baseline configuration of OsloCTM3 includes updates to the scavenging assumptions based on
previous studies investigating reasons for the high-altitude discrepancies (e.g., [*Hodnebrog et al.*,
2014; *Lund et al.*, 2017]. At high northern and southern latitudes, the model underestimates the
observed vertical profiles in the baseline. Increasing the model resolution does not have any
notable impact on the vertical profiles. There is a notable increase in high-latitude concentrations
when large-scale ice cloud scavenging is decreased. However, there is a simultaneous exacerbation
of model performance in the other latitude bands, pointing to potential tradeoffs when tuning
global parameters, as also illustrated by *Lund et al.* [2017]. Due to the significant altitude
dependence of the radiative effect of BC (e.g., [*Samset et al.*, 2013]), high altitude overestimations
will contribute to uncertainties in BC RFari. We also note that HIPPO3 was conducted in
March/April: Comparison with aircraft measurements from other seasons show a smaller
underestimation at high latitudes [*Lund et al.*, 2018].

3.3 Pre-industrial to present-day aerosols

With confidence in the model ability to reasonably represent current aerosol distributions
established, we next present an updated historical evolution of anthropogenic aerosols from pre-
industrial to present-day, and the consequent direct radiative effect (RFari) (Sect. 2.4). Figure 7
shows the net change in total aerosol load from 1750 to 2014. Full times series by species are given
in Table S4. To keep in line with the terminology used in the IPCC AR5, we now separate out
biomass burning BC and POA in a separate species "biomass". We also note that only the fine
mode fraction of nitrate contributes to the RFari and is inlcuded in Fig. 7. To illustrate the
contributions from additional emissions during the past 14 years, we also include the 2000-1750
difference. The values from the present study are also compared with results from the AeroCom
II models [*Myhre et al.*, 2013a], where emissions over the period 1850 to 2000 from *Lamarque et*
*al.* [2010] were used.

The most notable difference compared to the AeroCom II results is seen for biomass aerosols.
Biomass burning emissions have high interannual variability and this affects the analysis. While
the 1750-2014 difference is 0.23 mg m$^{-2}$, taking the difference between year 1750 and 2000 (black
triangle) results in a net change of only 0.03 mg m$^{-2}$. There is also a much larger change in the
burden of biomass aerosols in the AeroCom experiments, reflecting a more than 100% higher
emissions in 2000 compared to 1850 *Lamarque et al.* [2010] inventory. However, biomass aerosols
comprises both scattering OA and absorbing BC and, as seen below, these nearly cancel in terms
of RFari. Changes in sulfate and OA from pre-industrial to 2000 are slightly higher in the present
analysis than in AeroCom II, and the influence of additional emissions since 2000 is seen. The
OsloCTM3 is well below the AeroCom multi-model mean for nitrate. The OsloCTM2 was found
to be in the low range, but the multi-model was also influenced by some models giving high
estimates [*Myhre et al.*, 2013a]
Using the CEDSv16 emissions, we estimate a net RFari from all anthropogenic aerosols in 2014
relative to 1750 of -0.17 W m$^{-2}$. The RFari from sulfate is -0.30 W m$^{-2}$, while the contributions
from OA (combined fossil fuel plus biofuel POA and SOA), nitrate and biomass aerosols  are
smaller in magnitiude of -0.09, -0.02 and -0.0004 W m$^{-2}$, respectively. The RFari due to fossil fuel
and biofuel BC over the period is 0.31 W m$^{-2}$.
Figure 8a shows the time series of RFari by component, as well as the net, in the present analysis
(solid lines), and corresponding results reported in the IPCC AR5 (dashed lines). The net RFari
over time is mainly determined by the relative importance of compensating BC and sulfate RFari.
The most rapid increase in BC RFari is seen between 1950 and 1990, as emissions in Asia started
to grow, outweighing reductions in North America and Europe [*Hoesly et al.*, 2018]. After a period
of little change between 1990 and 2000, the rate of change increases again over the past two
decades, following strong emission increases in Asia and South Africa. Similarly, cooling
contribution from sulfate aerosols strenghtened from around mid-century. However, in contrast to
BC, the evolution is fairly flat after 1990. The last 20 years has seen a continuous reduction in
sulfur dioxide (SO$_2$) emissions in Europe, from around 30 to 5 Tg yr$^{-1}$ in CEDSv16, with a similar
trend in North America. While emissions in China continue to increase well into the 2000s, a
stabilization is seen after 2010, following introduction of stricter emission limits as part of a
program to desulfurize power plants [*Klimont et al.*, 2013]. During the same period, emissions in
India have risen. However, the net global SO$_2$ emission trend over the past few years is a slight
decline [*Hoesly et al.*, 2018]. This development is reflected in the net RFari, which reaches its peak
(i.e., strongest negative value) around 1990 and gradually becomes weaker thereafter. This trend
is more pronounced in the present analysis that in the IPCC AR5 estimates, where the forcing due
to sulfate is more flat in recent decades, suggesting that projected emission estimates
underestimated recent decreases in SO$_2$. The minimum net RFari value is also reached later in the
latter. Moreover, a recent study suggests that current inventories underestimate the decline in
Chinese SO$_2$ emissions and estimate a 75% reduction since 2007 [*Li et al.*, 2017]. In this case, the
weakening trend could be even stronger than estimated here. The insert in Fig. 8a focuses on recent
estimates of total RFari over the period 1990-2015. Using the ECLv5 emission inventory, *Myhre*
*et al.* [2017] found a global mean RFari due to changes in aerosol abundances over the period
1990-2015 of 0.05 ($\pm$0.04) W m$^{-2}$. Our results using CEDSv16 emissions are in close agreement
with these findings.

Over the past decades, there has been shift in emissions, from North America and Europe to South
and East Asia. This is also reflected in the zonally averaged net RFari over time in Fig. 8b. RFari
declined in magnitude north of 40°N after 1980, with particularly large year-to-year decreases
between 1990 and 1995, and from 2005 to 2010, and strengthened in magnitude between 10°-
30°N. The RFari also strengthened in the Southern Hemisphere subtropical region, reflecting
incresing emission in Africa and South America after 1970. However, the peak net RFari is
considerably weaker in 2014 than the peak in 1980. This mainly is due to fact that simultaneously
with the southwards shift, the sulfate burden has declined, while the BC burden has increased
steadily at the same latitudes, resulting in a weaker net RF. The past decade, the net RFari has
switched from negative to positive north of 70°N, due to a combination of stronger positive RF of
BC and from biomass burning aerosols.

Table S5 shows changes in burden, AOD, AAOD, RFari, and normalized RF over the period 1750-
2010 for individual aerosol components and the net RFari. Compared to earlier versions of
OsloCTM [*Myhre et al.*, 2009; *Myhre et al.*, 2013a] the normalized RF with respect to AOD is
lower because of short lifetime of BC resulting in smaller abundance of BC above clouds, whereas
normalized RF with burden is comparable to earlier estimates because of higher MAC compensate
for short lifetime of BC. Weaker normalized RF of OA (POA and SOA) than earlier OsloCTM
versions is due to the inclusion of absorbing OA.
In the present study we have used an updated parameterization of BC absorption based on *Zanatta*
*et al.* [2016] (Sect. 2.4), which takes into account the ratio of non-BC-to-BC material and results
in a MAC of 12.5 m$^2$ g$^{-1}$ at 550 nm. This is 26% higher than the 9.94 m$^2$ g$^{-1}$ using the approach
from *Bond and Bergstrom* [2006]. Using the latter, we estimate a BC RFari in 2014 relative to
1750 of 0.23 W m$^{-2}$, 25% lower than the 0.31 W m$^{-2}$ calculated based on *Zanatta et al.* [2016].
These results emphasize the importance of assumptions and uncertainties related to the BC
absorption.
The magnitude of RFari by scattering aerosols is sensitive to assumptions about absorption by
organic aerosols, so-called brown carbon (BrC). Observational studies have provided evidence for
the existence of such particles, and modeling studies suggest they could be responsible for a
substantial fraction of total aerosol absorption, although the spread in estimates is wide (e.g., *Feng*
*et al.* [2013] and reference therein). In the present study we assume a considerable fraction of
absorption by OA (Sect. 2.4). Assuming purely scattering aerosols, the RFari from OA is  -0.13W
m$^{-2}$; acounting for BrC absorption this is weakened to -0.09 W m$^{-2}$. Splitting total OA RFari into
contributions from primary and secondary aerosols, we find that purely scattering POA gives a
RFari of -0.07 W m$^{-2}$ compared to -0.06 Wm$^{-2}$ with absorption. The corresponding numbers for
SOA are -0.06 and -0.03 W m$^{-2}$. This indicates that in OsloCTM3, the absorbing properties of SOA
are relatively more important than for POA. This is likely due to the generally higher altitude of
SOA than POA (Fig. S8) in combination with the increasing radiative efficiency of absorbing
aerosols with altitude [*Samset et al.*, 2013]. However, due to the weaker overall contributions from
OA, our results indicate that differences in parameterization of BC absorption can be more
important than uncertainties in absorption by BrC for the net RFari.

4 Discussion

Our estimate of total net RFari in 2014 relative to 1750 is weaker in magnitude than the best
estimate for the 1750-2010 period reported by IPCC AR5. The difference is due to a combination
of factors, including weaker contributions from both cooling aerosols and BC. Despite
considerably higher BC emissions in the CEDSv16 inventory compared to older inventories, we
calculate a weaker BC RFari than reported in AR5, hence going in the opposite direction of
explaining the difference to IPCC AR5 total RFari. The IPCC AR5 best estimate for fossil fuel
and biofuel BC of 0.4 (0.05 to 0.8) W m$^{-2}$ [*Boucher et al.*, 2013] was based mainly on the two
studies by *Myhre et al.* [2013a] and *Bond et al.* [2013], who derived estimates of BC RFari of 0.23
(0.06 to 0.48) W m$^{-2}$ and 0.51 (0.06 to 0.91) W m$^{-2}$, respectively. The spread between the two is
largely attributed to methodological differences: *Bond et al.* [2013] used an observationally
weighted scaling of results to match those based on AERONET AAOD, which was not adopted
by *Myhre et al.* [2013a]. Such ad-hoc adjustments typically result in higher estimates (*Wang et al.*
[2018] and references therein). Moreover, a recent study by *Wang et al.* [2018] suggest that
representativeness error arising when constraining coarse resolution models with AERONET
AAOD could result in a 30% overestimation of BC RFari, which explains some of the differences
between bottom-up and observationally constrained numbers. The BC RFari estimate from the
present study is around 20% higher than the AeroCom multi-model mean from *Myhre et al.* [2013a]
when calculated over the same period 1850-2000. This reflects the higher emissions in the
CEDSv16 emission inventory than in *Lamarque et al.* [2010], as well as a higher MAC.

A significant range from -0.85 to +0.15 W m$^{-2}$ surrounds the central RFari estimate of -0.35 W m$^{-2}$
from IPCC AR5 [*Boucher et al.*, 2013], caused by the large spread in underlying simulated
aerosol distributions. Deficiencies in the ability of global models to reproduce atmospheric aerosol
concentrations can propagate to uncertainties in RF estimates. As shown in Sect. 3, the OsloCTM3
generally lies close to or above the multi-model mean of anthropogenic aerosol burdens from
recent studies and is found to perform reasonably well compared with observations and other
global models, with improvements over the predecessor OsloCTM2. In particular, recent progress
towards constraining the vertical distribution of BC concentrations has resulted in improved
agreement between modeled and observed vertical BC profiles over the Pacific Ocean with less of
the high-altitude overestimation seen in earlier studies. However, as shown by *Lund et al.* [2018],
there are discrepancies compared to recent aircraft measurements over the Atlantic Ocean. A
remaining challenge is the model underestimation of Arctic BC concentrations. However, this is
seen mainly during winter and early spring, when the direct aerosol effect is small due to lack of
sunlight. In contrast, the higher emissions in the CEDSv16 inventory also results in an improved
agreement with BC surface concentrations over Asia.
In general, we find lower surface sulfate concentrations in the model compared with measurements.
This could contribute to an underestimation of the sulfate RFari, which is weaker in the present
study than in IPCC AR5. An underestimation of observed AOD in Asia is also found, however,
the implication of this bias on RF is not straightforward to assess, as it is complicated by the mix
of absorbing and scattering aerosols. We note that the global mean sulfate burden is higher in the
OsloCTM3 than in most of the global models participating in the AeroCom III experiment (Sect.
3.1, *Bian et al.* [2017]), and that the OsloCTM3 performs similarly to or better than other AeroCom
Phase III models in terms of nitrate and sulfate surface concentrations, at least for measurements
from CASTNET [*Bian et al.*, 2017]. Nevertheless, the model diversity in simulated nitrate and
sulfate remains large and, although all models capture the main observed features in concentrations,
further work is needed to resolve the differences and improve model performance for these species.
While a comprehensive quantitive uncertainty analysis of the updated RFari estimate is not
possible within the scope of this study, we explore the order of magnitude uncertainties due to
"internal" factors such as scavenging parameterizations and model resolution by performing
sensitivity tests. Changes in global burden on the order of 10-20%, and up to 65%, were found
(Sect. 3.2). However, compared to observations of surface concentrations in near-source regions,
total AOD and vertical distribution of BC concentrations, we saw that the model generally
performed the best in its baseline configuration. Furthermore, the largest changes in the simulated
AOD and aerosol distributions were found in high-latitude regions, whereas changes over land
where the concentrations, and hence subsequent RF is localized, were smaller. For certain regions
and observables, there were notable differences between the baseline and sensitivity simulations.
For instance, an improvemet in the baseline compared to using the CMIP5 emission inventory was
seen for BC surface concentrations, in particular in Asia, while the NMB of AOD compared to
AERONET stations in the same region was reduced in the simulation with higher spatial resolution.
The importance of using the correct meteorological year was also seen. Such uncertainties will
translate to the RFari estimates, along with uncertainties in optical properties such as absorption
by organic aerosols and parameterization of BC absorption (Sect. 3.3).
Estimates of radiative impacts depend critically on the confidence in the emission inventories. A
detailed discussion of uncertainties in the CEDS inventory is provided by *Hoesly et al.* [2018]. On
a global level, the uncertainty in $SO_2$ emissions tend to be relatively low, although there is an
indication of missing $SO_2$ sources in particular in the Persian Gulf [*McLinden et al.*, 2016], whereas
emission factors for BC, OC, NOx, CO and VOCs have higher uncertainties. Uncertainties in
country-specific emissions can also be much larger, which is particularly true for carbonaceous
aerosols. In future CEDS versions, a quantitative uncertainty analysis is planned [*Hoesly et al.*,
2018], which will provide valuble input to modeling studies.
Our study does not include anthropogenic dust, i.e., wind-blown dust from soils disturbed by
human activities such as land use practices, deforestation and agriculture, and fugitive combustion
and industrial dust from urban sources. These sources could contribute an important fraction of
emissions and ambient $PM_{2.5}$ concentrations in some regions [*Paul et al.*, 2012; *Sajeev et al.*, 2017],
but are missing from most models today. For instance, a recent study found a 2–16 mg m$^{-3}$ increase
in PM2.5 concentrations in East and South Asia from anthropogenic fugitive, combustion, and
industrial dust emissions. However, the transport processes and optical properties, and hence,
radiative impact, is poorly known. We also do not include the effect of aerosol-cloud interactions,
which are crucial for the net climate impact of aerosols. For instance, recent studies suggest that
the impact of BC on global temperature response is small due to largely compensating direct and
rapid adjustment effects [*Samset and Myhre*, 2015; *Stjern et al.*, 2017]. The composition and
distribution of aerosols and oxidants in the pre-industrial atmosphere is uncertain and poorly
constrained by observations. However, while this is an important source of uncertainty in estimates
of RF due to aerosol-induced cloud albedo changes, it is less important for RFari because the
forcing scales quite linearly with aerosol burden [*Carslaw et al.*, 2017].
5 Conclusions
In this study, we have documented the third generation of the Oslo chemical transport model
(OsloCTM3) and evaluated the simulated distributions of aerosols, including results from a range
of simulations to investigate the sensitivity to uncertainties in scavenging processes, input of
emissions and meteorological data and resolution. We have then used the new historical CEDS
emission inventory (version 2016; CEDSv16), which will also be used in the upcoming CMIP6,
to simulate the temporal evolution of atmospheric concentrations of anthropogenic aerosols, and
quantified the temporal evolution of the subsequent radiative forcing due to aerosol-radiation
interactions (RFari).
The total AOD from the OsloCTM3 is in good agreement with observations from the AERONET
network with a correlation of 0.82 and a normalized mean bias (NMB) of -11.8%. Regionally, the
underestimation of observed AOD is higher for stations in China and India than in Europe and
North America, as also reflected from the comparison against measured aerosol surface
concentrations. High correlations 0.80-0.90 are also found for surface concentrations of BC, OC,
sulfate and nitrate aerosols compared with all measurements across Europe, North America and
Asia. The corresponding NMB range from -23% for BC and nitrate to -46% and -52% for OC and
sulfate, respectively. The OsloCTM3 performs notably better than its predecessor OsloCTM2 in
terms of high-altitude BC distribution as compared with observed BC concentration profiles over
the Pacific Ocean from the HIPPO3 campaign. In constrast, the model continues to underestimate
observed surface levels of BC during winter and spring. Compared with other recent estimates of
aerosol burdens, the OsloCTM3 generally lies close to or above the mean of other global models.
Increasing or reducing the scavenging efficiency, moving to a finer resolution, and using the wrong
meteorological year or a different emission inventory results in changes in the global mean aerosol
burdens of up to 65%. The burdens of BC, OC and sulfate are particularly sensitive to a reduced
efficiency of removal by large-scale ice clouds; a 10 percentage point reduction increases the
global burden by 40%, 65% and 20%, respectively. A corresponding increase in the efficiency
gives around 10% lower burdens. A significantly better agreement with BC surface concentrations
is found when using the CEDSv16 emission inventory compared with the RCP4.5. Furthermore,
a notable reduction in the bias of AOD compared to AERONET observations in Asia is found
when increasing the horizontal resolution, while the correlation is reduced when using the wrong
meteorological year. However, we find no clear evidence of consistently better model performance
across all observables and regions in the sensitivity tests than in the baseline configuration. This
may in part be influenced by the geographical coverage of observations, as the largest differences
in concentrations and AOD from the baseline is found at high altitudes and latitudes where the
availability of constraining measurements is limited.
Using the CEDSv16 historical emission inventory we estimate a total net RFari from all
anthropogenic aerosols, relative to 1750, of -0.17 W m$^{-2}$. This is significantly weaker than the best
estimate reported in the IPCC AR5, due to a combination of factors resulting in weaker
contributions from both cooling aerosols and BC in our simulations. Our updated RFari estimate
is based on a single global model. As shown by previous studies, there is a large spread estimates
of RFari due to the spread in modeled aerosol distributions. The present analysis shows that
uncertainties in emissions, scavenging and optical properties of aerosols can have important
impacts on the simulated AOD and subsequent forcing estimates within one model. Additional
studies to place our estimates in the context of multi-model spread and provide a comprehensive
uncertainty analysis are needed ahead of the IPCC Sixth Assessment Report.


Data availability
The CEDS anthropogenic emissions data is published within the ESGF system https://esgf-
node.llnl.gov/search/input4mips/. Surface observations used in this study are collected from the
following publicly available databases: the EBAS database (http://ebas.nilu.no/) hosted by NILU
– Norwegian Institute for Air Research. The US national Clean Air Status and Trends monitoring
network (CASTNET), available at http://www.epa.gov/castnet. The Interagency Monitoring of
Protected Visual Environments (IMPROVE), a collaborative association of state, tribal, and
federal agencies, and international partners, with the US EPA as the primary funding source and
support from the National Park Service. Data available from

http://vista.cira.colostate.edu/Improve/. MODIS and MISR AOD retrievals are downloaded from https://giovanni.gsfc.nasa.gov/giovanni/. Aircraft measurements from the HIPPO3 flights available from https://www.eol.ucar.edu/node/524. The modeled and measured aerosol surface concentrations used in the model evaluation are publicly available via the ACTRiS data center (https://doi.org/10.21336/GEN.3). Remaining model output available upon request from Marianne T. Lund (m.t.lund@cicero.oslo.no).

Code availability

The OsloCTM3 is stored in a SVN repository at the University of Oslo central subversion system and is available upon request. Please contact m.t.lund@cicero.oslo.no. In this paper, we use the official version 1.0, OsloCTM3v1.0.

Acknowledgements

MTL, GUM, AHS, RBS acknowledges funding from the Norwegian Research Council through grants 250573 (SUPER) and 248834 (QUISARC). The National Center for Atmospheric Research (NCAR) is sponsored by the National Science Foundation (NSF). The authors also acknowledge funding of the Horizon 2020 research and innovation programme ACTRIS-2 Integrating Activities (IA) (grant agreement No 654109). The AeroCom database is maintained through basic funding from the Norwegian Meteorological Institute. We would like to express our thanks to all those who are involved in AERONET, IMPROVE, CASTNET, EMEP and ACTRIS measurements efforts and have contributed through operating sites, performing chemical analysis and by submissions of data to public data bases. We also acknowledge the Research Council of Norway's programme for supercomputing (NOTUR). Thanks to Richard Rud (NILU ATMOS) for assistance with data availability through the ACTRiS data center.

Competing interests

The authors declare that they have no conflict of interest.

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

Tables

*Table 1: Summary and description of simulations in this study*

| Name | Athropogenic emissions | Year | Res | Description |
|------|------------------------|------|-----|-------------|
| CEDSv16/CMIP6 | CEDS, version released in 2016 | 2010 | 2x2 | Baseline simulation, 2.25x2.25 degree resolution |
| ECLv5 | ECLIPSEv5 | 2010 | 2x2 | As baseline, but with ECLIPSEv5 emissions |
| RCP/CMIP5 | RCP4.5 | 2010 | 2x2 | As baseline, but RCP4.5/CMIP5 emissions |
| LSIDEC | CEDS | 2010 | 2x2 | Reduced scavenging of all aerosols by large-scale ice clouds |
| LSIINC | CEDS | 2010 | 2x2 | Increased scavenging of all aerosols by large-scale ice clouds |
| SOLDEC | CEDS | 2010 | 2x2 | Decreased scavenging of all aerosols by convective and large-scale liquid preciptation |
| 1x1RES | CEDS | 2010 | 1x1 | Same as baseline, but on 1.125x1.125 degree resolution |
| METDTA | CEDS | 2010 | 2x2 | Year 2010 emissions, but 2000 meteorology |
| Historical | CEDS/ | 1750-2014 | 2x2 | Time-slice simulations for year 1750, 1850, 1900, 1910, 1920, 1930, 1940, 1950, 1960, 1970, 1980, 1985, 1990, 1995, 2000, 2005, 2010, 2014 |


*Table 2: Fraction of aerosol mass available for wet scavenging by convective, large-scale liquid*
*and large-scale ice precipitation in baseline setup and in the three sensitivity tests.*
*Phil=hydrophilic, phob=hydrophobic.*

| Simulation | Precipitation type | Sulfate | OM phil | OM phob | BC phil | BC Phob | Nitrate | SOA | Sea salt | Dust |
|------------|-------------------|---------|---------|---------|---------|---------|---------|-----|----------|------|
| CEDSv16/ CMIP6 | Convective | 1 | 1 | 1 | 1 | 1 | 1 | 0.8 | 1 | 1 |
| | LS-liquid | 1 | 1 | 0 | 1 | 0 | 1 | 0.8 | 1 | 1 |
| | LS-ice | 0.1 | 0.1 | 0.2 | 0.1 | 0.2 | 0.1 | 0.16 | 0.1 | 0.5 |
| LSIINC | LS-ice | 0.3 | 0.3 | 0.4 | 0.3 | 0.4 | 0.3 | 0.32 | 0.3 | 0.7 |
| LSIDEC | LS-ice | 0.001 | 0.001 | 0.1 | 0.001 | 0.1 | 0.001 | 0.001 | 0.001 | 0.1 |
| SOLDEC | Convective | 0.8 | 0.8 | 0.8 | 0.8 | 0.8 | 0.8 | 0.6 | 0.8 | 0.8 |
| | LS-liquid | 0.8 | 0.8 | 0 | 0.8 | 0 | 0.8 | 0.6 | 0.8 | 0.8 |


*Table 3: Global, annual mean aerosol burdens [mg m$^{-2}$] and total AOD in the baseline and sensitivity simulations. Parentheses in the top row give the atmospheric residence time (ratio of burden to total wet plus dry scavenging) [days]. Corresponding values for the sensitivity simulations are given in Table S3.*

| Simulation | BC | OA | Sulfate | NH4 (fine+coarse) | Nitrate (fine) | Nitrate (coarse) | Sea salt | Dust | AOD |
|---|---|---|---|---|---|---|---|---|---|
| **CEDS/CMIP6** | **0.23** **(4.4)** | **3.4**[§] **(5.3)** | **5.4** **(5.4)** | **0.68** **(3.5)** | **0.17** **(4.2)** | **3.9** **(5.2)** | **12** **(0.46)** | **39** **(3.4)** | **0.13** |
| ECLv5 | 0.21 | 3.1 | 5.1 | 0.65 | 0.15 | 3.7 | 12 | 39 | 0.13 |
| RCP/CMIP5 | 0.18 | 3.2 | 5.3 | 0.63 | 0.13 | 3.7 | 12 | 39 | 0.13 |
| LSIINC | 0.21 | 2.8 | 4.9 | 0.63 | 0.17 | 3.4 | 11 | 39 | 0.12 |
| LSIDEC | 0.32 | 5.3 | 6.5 | 0.79 | 0.16 | 4.7 | 14 | 43 | 0.16 |
| SOLDEC | 0.26 | 3.6 | 6.1 | 0.78 | 0.16 | 5.2 | 15 | 42 | 0.15 |
| 1x1RES | 0.24 | 3.4 | 5.6 | 0.71 | 0.19 | 3.6 | 12 | 38 | 0.14 |
| METDTA | 0.22 | 3.0 | 5.5 | 0.69 | 0.16 | 3.8 | 12 | 42 | 0.13 |

[§] SOA: 1.1 mg m$^{-2}$ [5.8 days] and POA: 2.3 mg m$^{-2}$ [5.1 days]

Figures

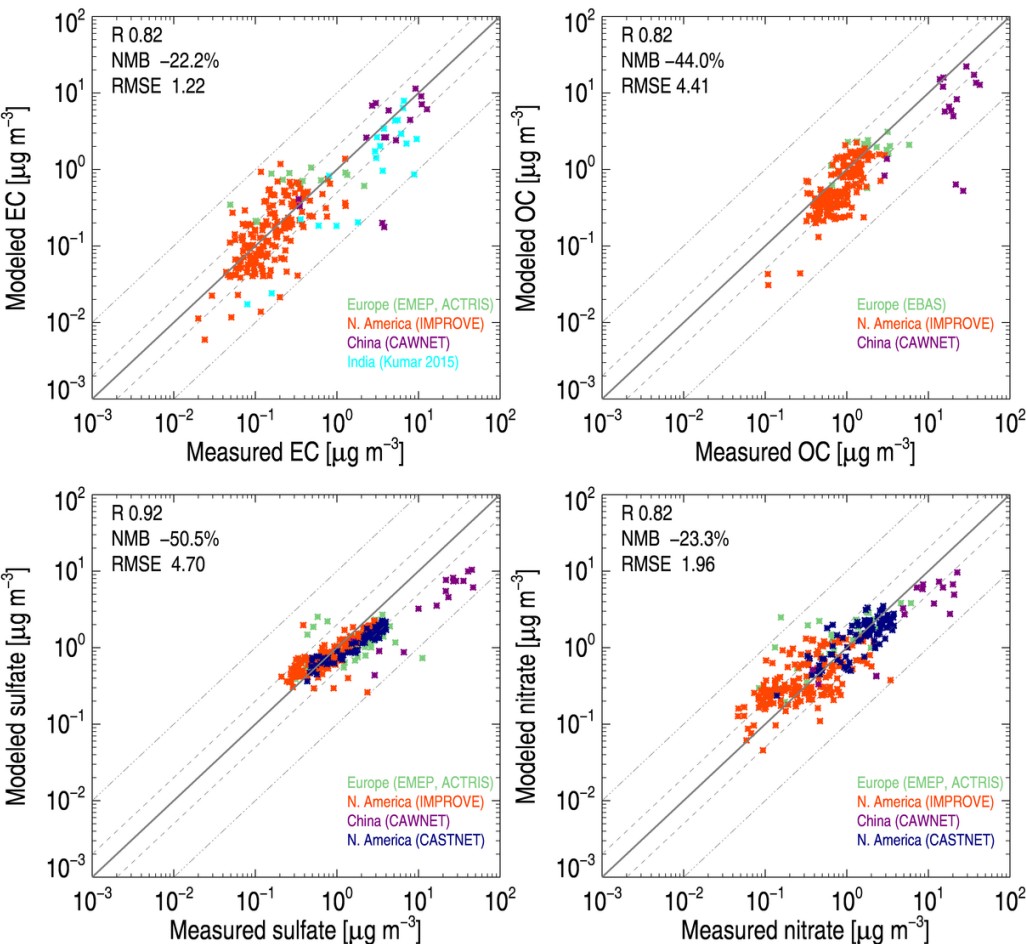



*Figure 1: Annual mean modeled versus measured aerosol surface concentrations of a) EC, b)*
*OC, c) sulfate and d) nitrate from the IMPROVE, EMEP, ACTRIS, CASTNET and CAWNET*
*measurements networks.*

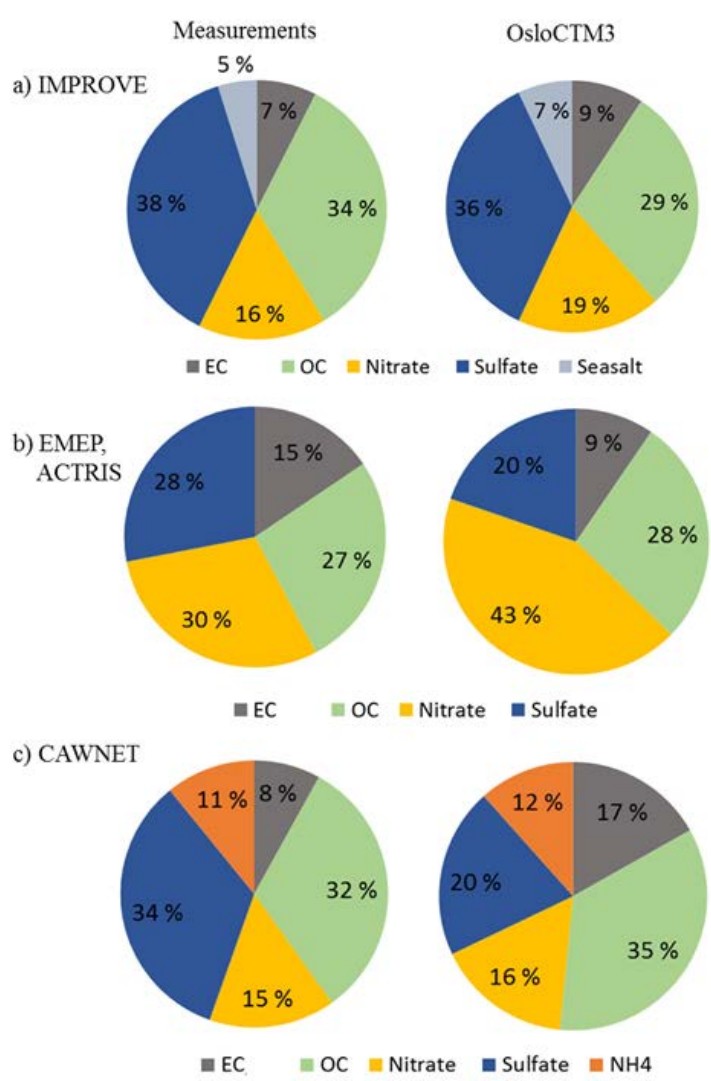



*Figure 2: Aerosol composition (fraction of total aerosol mass) derived from the IMPROVE, EMEP,*
*ACTRIS and CAWNET networks (left column) and corresponding OsloCTM3 results (right*
*column).*


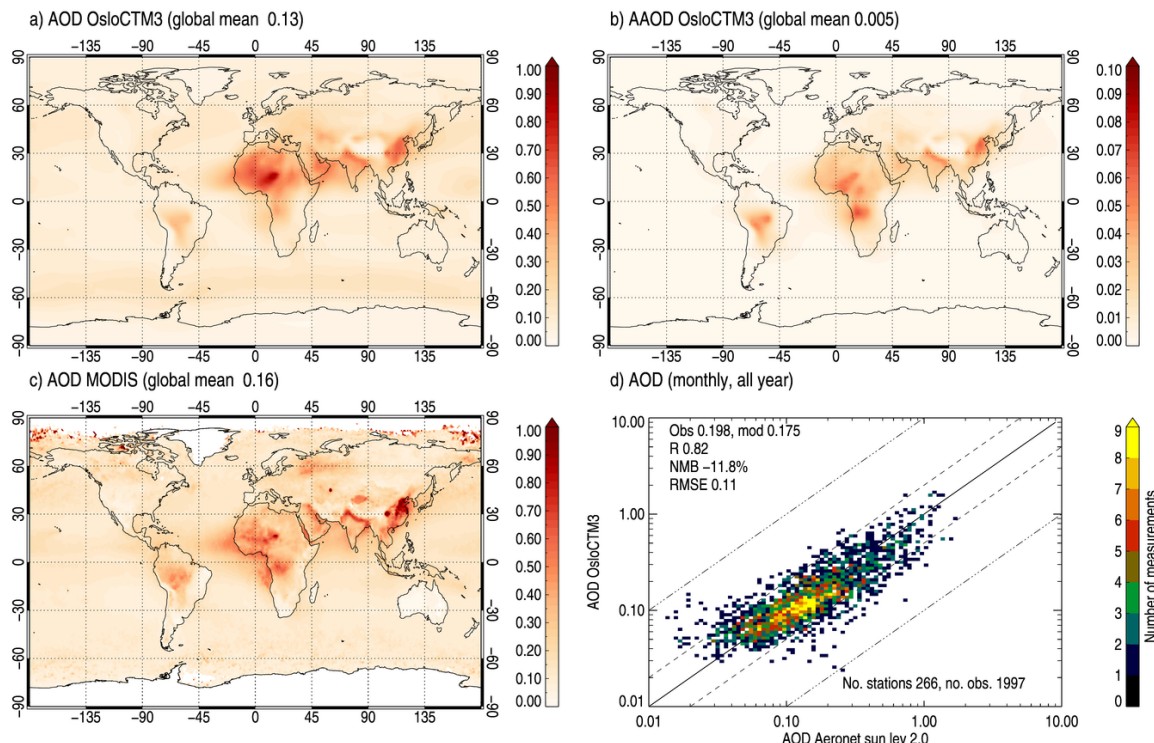


*Figure 3: Annual mean (year 2010) modeled a) AOD and b) AAOD, c) MODIS-Aqua AOD*
*retrieval and d) scatter density plot of comparison of simulated AOD against monthly mean*
*AERONET observations.*



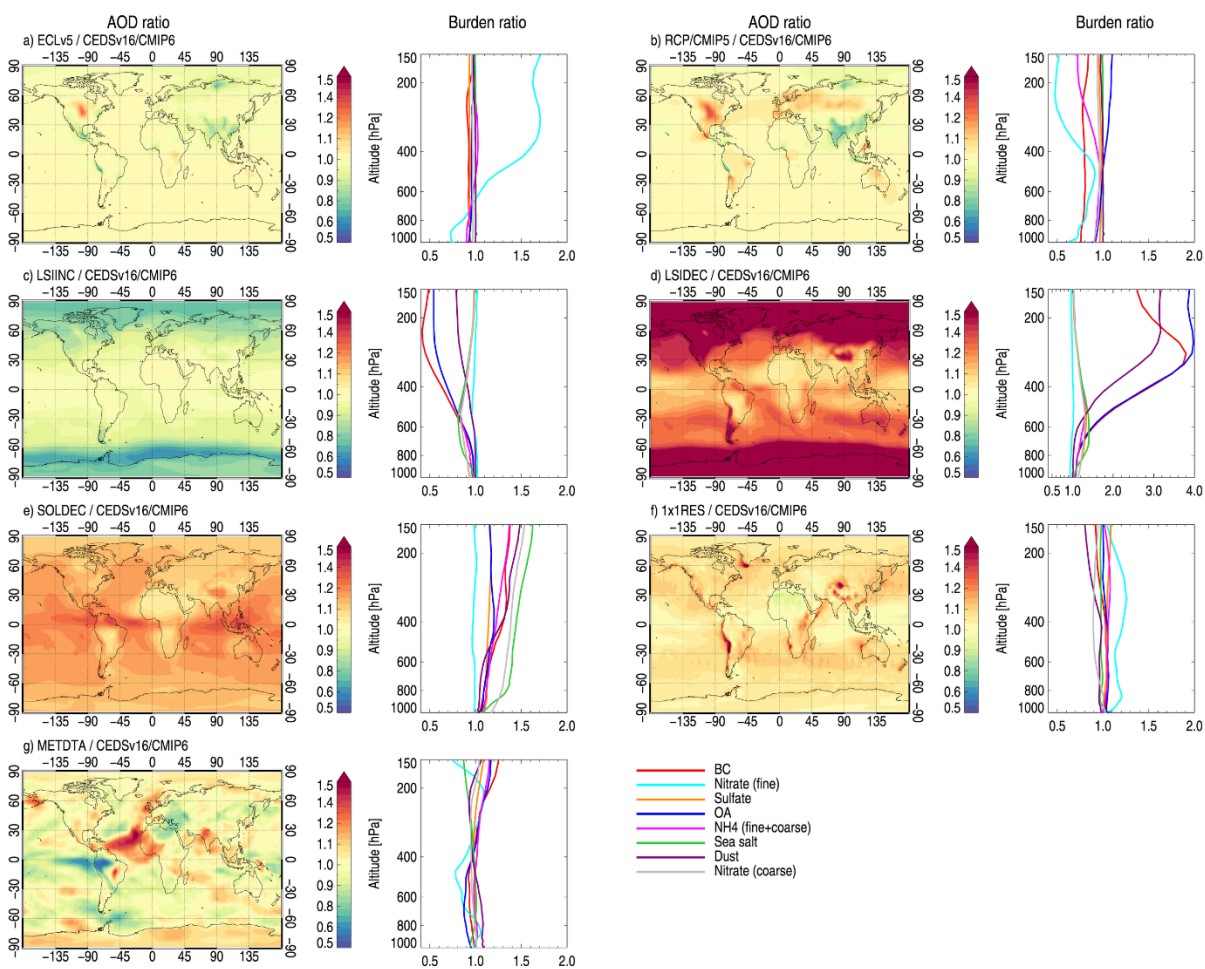


*Figure 4: Ratio of each sensitivity simulation relative to the baseline for AOD (columns 1 and 3)*
*and total burden by species in each model layer (columns 2 and 4).*


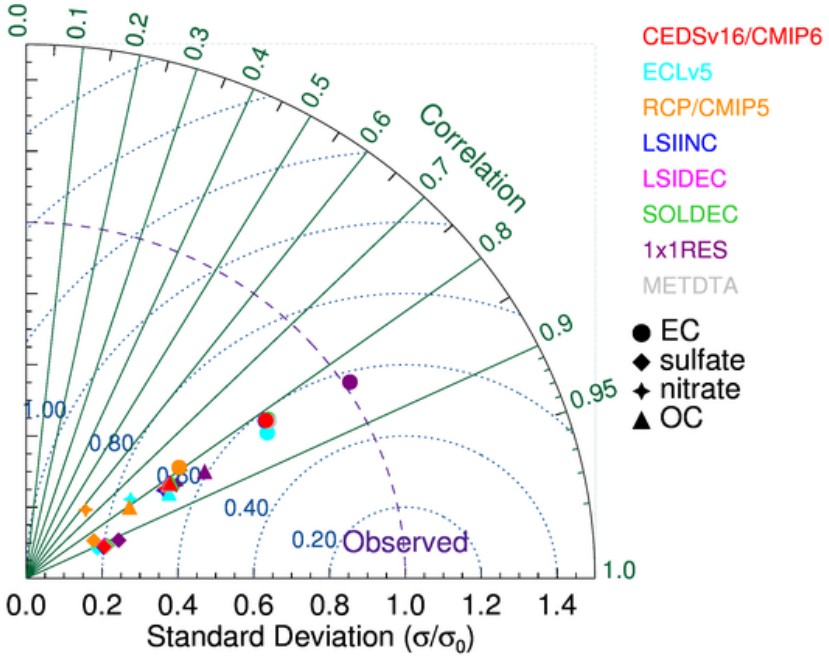


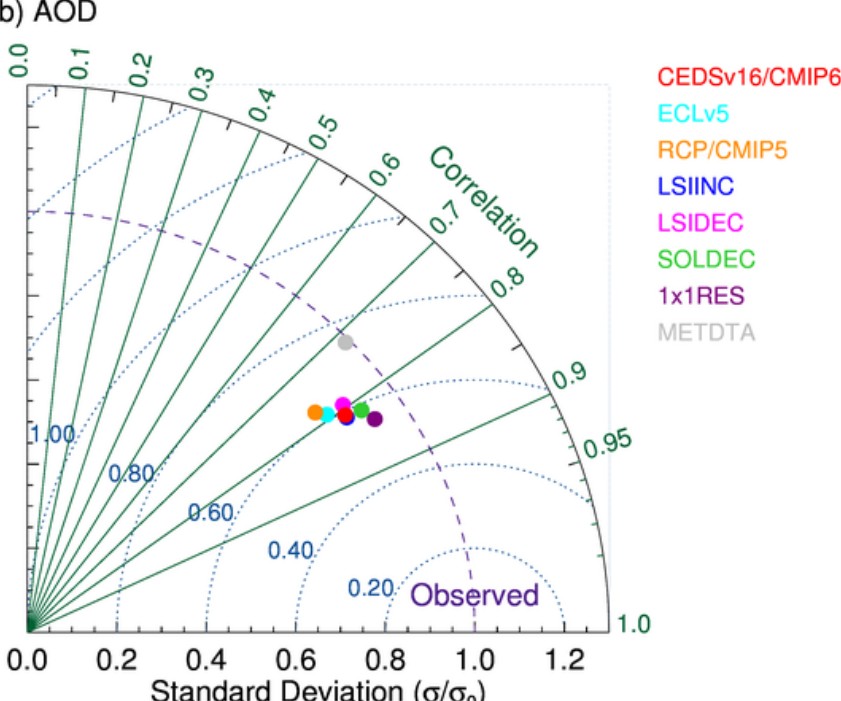


*Figure 5: Taylor diagram of modeled and measured aerosol surface concentrations in the baseline simulation and sensitivity tests using all observations in Fig. 1.*




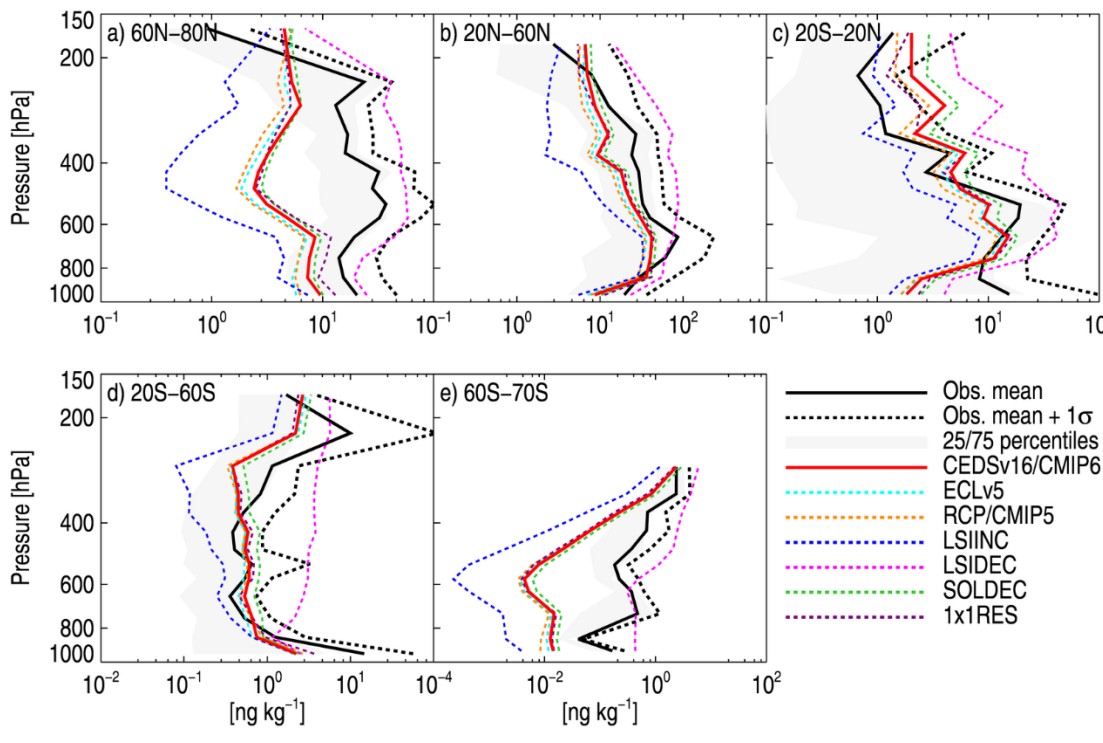

*Figure 6: Modeled vertical BC profiles against rBC aircraft measurements in five different*
*latitudes bands over the Pacific Ocean from the HIPPO3 flight campaign. Model data is*
*extracted along the flight track using an online flight simulator. Black lines: mean of*
*observations (solid), mean + plus 1 standard deviation (dashed). Colored lines: OsloCTM3*
*baseline (CEDSv16/CMIP6) (solid), sensitivity simulations (dashed).*

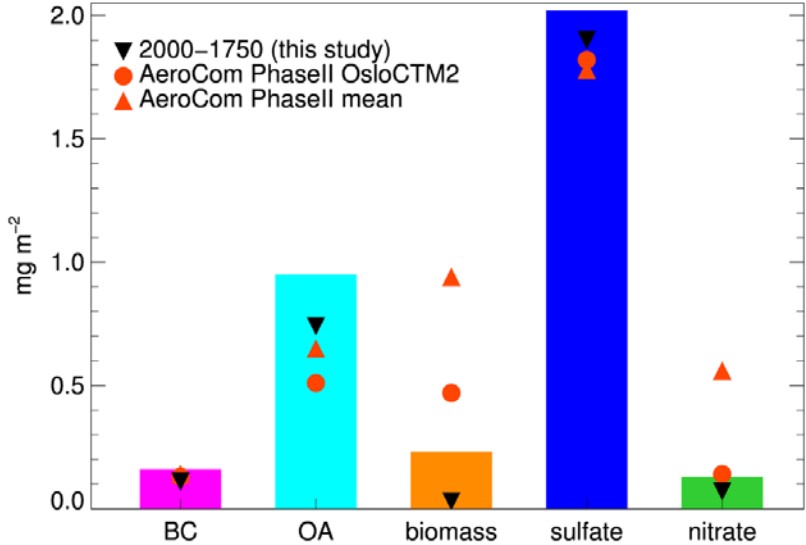


*Figure 7: Change in anthropogenic aerosol load over the period 1750 to 2014 using CEDSv16*
*emissions. Black symbols show the 1750 to 2000 difference and red symbols show multi-model*
*mean and OsloCTM2 results from the AeroCom II experiments [Myhre et al., 2013a].*

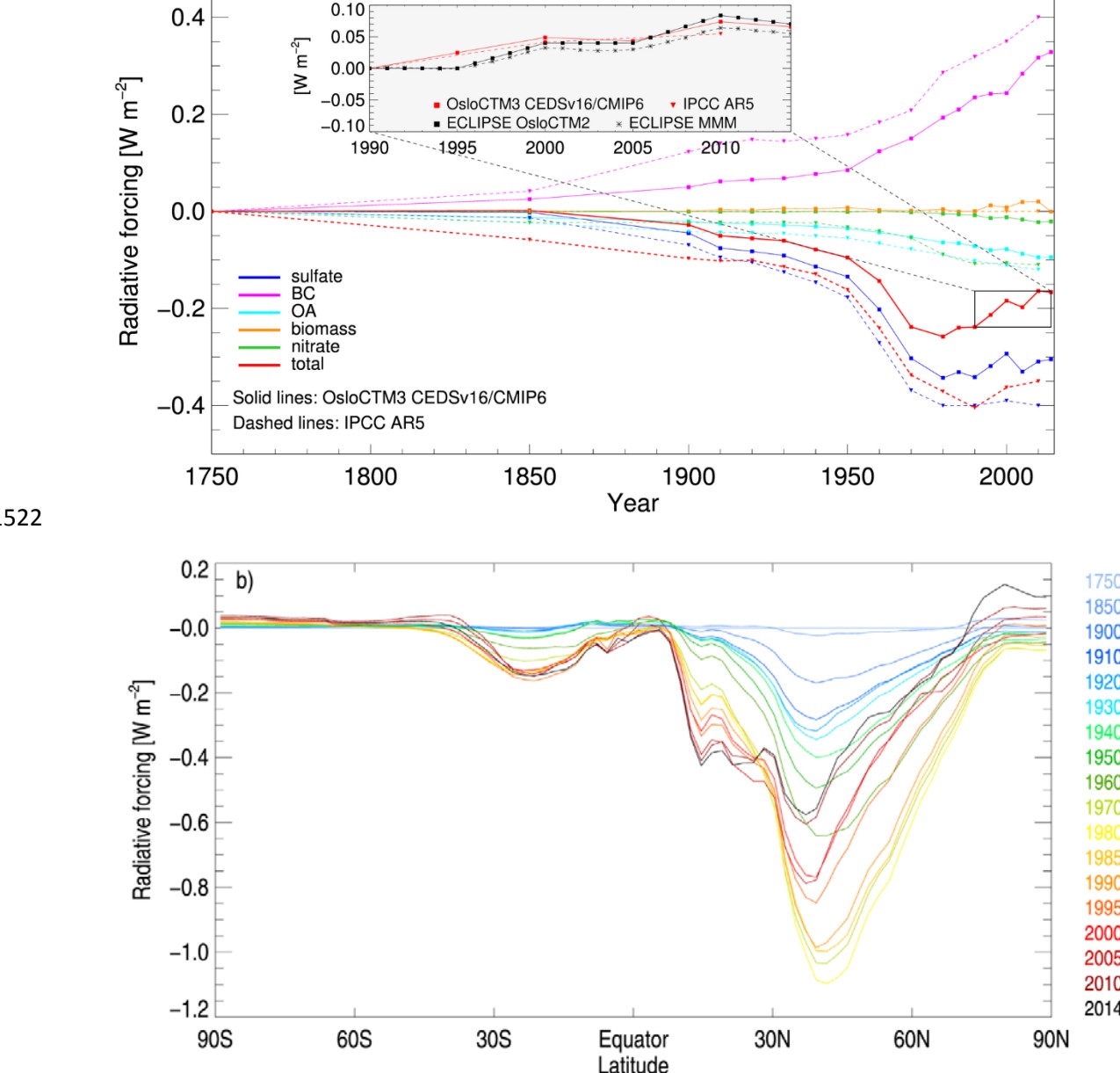



*Figure 8: a) Time evolution of RFari. Solid lines show OsloCTM3 results from the current study,*
*while dashed lines show results from IPCC AR5[Myhre et al., 2013b]. The inset shows the change*
*in total RFari between 1990 and 2015 in the current study compared with IPCC AR5 and multi-*
*model mean and OsloCTM2 results from Myhre et al. [2017] using ECLv5 emissions. b) zonal*
*mean RFari 1750-2014.*