# Peer review of "Concentrations and radiative forcing of anthropogenic aerosols from 1750-2014"

_Geoscientific Model Development, 2018_

## Short Comment (SC1) · 31 Jul 2018

Dear authors,

In my role as Executive editor of GMD, I would like to bring to your attention our Editorial version 1.1:

http://www.geosci-model-dev.net/8/3487/2015/gmd-8-3487-2015.html

This highlights some requirements of papers published in GMD, which is also available

on the GMD website in the 'Manuscript Types' section:

http://www.geoscientific-model-development.net/submission/manuscript_types.html

In particular, please note that for your paper, the following requirement has not been met in the Discussions paper:

- "All papers must include a section, at the end of the paper, entitled 'Code availability'. Here, either instructions for obtaining the code, or the reasons why the code is not available should be clearly stated. It is preferred for the code to be uploaded as a supplement or to be made available at a data repository with an associated DOI (digital object identifier) for the exact model version described in the paper. Alternatively, for established models, there may be an existing means of accessing the code through a particular system. In this case, there must exist a means of permanently accessing the precise model version described in the paper. In some cases, authors may prefer to put models on their own website, or to act as a point of contact for obtaining the code. Given the impermanence of websites and email addresses, this is not encouraged, and authors should consider improving the availability with a more permanent arrangement. After the paper is accepted the model archive should be updated to include a link to the GMD paper."

Thus please add a Code Availability Section and provide the required information how to access the exact Code Version of the OsloCTM3 used in your publication.

Yours,

Astrid Kerkweg
* * *

---

## Referee Comment (RC1) · Anonymous Referee #1 · 30 Aug 2018

General comments:

This paper evaluates the present-day aerosol distributions and radiative forcing in the latest OsloCTM3 chemical transport model. Surface concentrations, vertical distributions as well as aerosol optical depth are evaluated for the year 2010 against an extensive set of ground-based and satellite measurements. Uncertainties in the aerosol distributions are also assessed through a number of sensitivity studies assessing the uncertainty in the aerosol emissions, removal processes, meteorology and model resolution. The model is then used to assess the historical aerosol direct radiative forcing

(RF) from 1750-2010. The net radiative forcing as well as the RF due to individual species are compared against the most recent published literature and CMIP5 estimates. Aerosol-cloud interactions are not assessed in this study.

This is a well-written, clearly presented, model evaluation manuscript. This paper does not document the latest updates to the OsloCTM3 model, but appropriate references are provided for this. Instead it documents the latest aerosol simulations with this model using the latest CMIP6 emissions inventory and subsequently derives the latest direct aerosol RF estimate. So I believe it is within the GMD scope as a Model Evaluation paper. This paper is timely given the CMIP6 project is now well underway and will provide a useful update to direct aerosol RF estimates in this regard.

I recommend this paper to be published subject to a number of revisions I detail below.

Specific comments:

I struggled to appreciate the motivation for the additional sensitivity simulations conducted in this study. While these studies are useful and worth reporting the motivation for conducting these studies and link with the rest of the paper needs to be made clearer. This is probably most easily achieved in the Introduction.

In Sect 2.3 Please report the length of each time slice simulation and how this impacts the signal to noise in resulting RF estimates. Are you running 20-30 years for each time slice?

In Sect 3.3 please describe clearly either explicitly or through appropriate references how you calculated the RFari. Have you calculated an effective radiative forcing or used the more traditional radiative forcing metric? It is currently unclear. Also, how are the individual species RFari determined - through species only runs or is the RT code able to output this?. Please provide this detail in the manuscript.

Use of older version of CEDS emission inventory: I am not convinced that the historical RFari will not be impacted at all by the choice of CEDS emission inventory. Changes

in spatial distribution of emission could impact aerosol removal, transport and thus lifetime, temporal shifts in the distribution could also potentially impact the historical evolution of the ERF. While I understand the computational burden of repeating all tests, two runs using the new CEDS version with 1750 and for instance year 2010 emissions would allow you to quantify the impact on the RF fairly easily to allow you to justify it in the text. It looks like you may have the 2010 simulation already from you Fig S4 plots.

Furthermore limiting your evaluation to BC is also questionable as I would expect notable differences in for instance SO2. I would request that you extend this evaluation over the USA to all aerosol species where you have observations.

Changes to the large-scale ice scavenging efficiency have a large control on the global aerosol distribution. The paper would benefit from a more detailed description of how the LS ice (and liquid) scavenging is parameterized in the model in Sect 2.1

Technical corrections:

Line 70: 2011 - this should be 2010

Line 170: It would be worthwhile to report here what global scaling factor you have used in the Gantt parametersation of marine OM. Gantt et al. 2015 I think use a global scaling factor of 6 but this is believed to be highly model dependent.

Line 281: Explicity state the aerosol concentration threshold below which you apply the Bond and Bergstrom method. Why do you change approaches? Again state the motivation for these different approaches in different regimes and why they need to be made. Does Zanatta lead to too high a MAC in the low aerosol regimes? Has this been constrained by observations - if yes, provide appropriate reference here. Some discussion of the uncertainties in both approaches as this links to your uncertainties in BC RFari discussed in Sect 3.3

Line 329-330: It would be useful here to be clear on what model variables you've

sampled at 3 hour resolution and what ones you've sampled monthly and why you needed to do this.

Line 452: EBAS - I think this should refer to EMEP/ACTRIS. EBAS has not been previously mentioned.

Line 468: Remove "as for surface concentrations,"

Table 3: It would be useful to have the % change in burden listed alongside the burdens of all the sensitivity experiments. Perhaps in parentheses next to the burdens.

Line 541-544: Can you actually make this statement? You do not show any impact on surface concentrations. I do not think this paragraph adds any value to the manuscript and would remove.

Line 555: Could the higher correlation and lower bias found in the 1x1RES test not just be a consequence of improved spatial sampling in your comparison and not actually due to an improved distribution of AOD? As you do not evaluate the species whose emissions would depend more strongly on resolution (ie: dust, sea salt) it's hard to quantify the benefit here.

Line 638: OsloCTM3fast - what is this?? There is no prior mention of a fast configuration!!

Line 688: "These results emphasize the importance of assumptions related to the BC absorption" –> these results emphasize the importance of assumptions and uncertainties related to the BC absorption.

Figure 6: It would be more informative to show obs +/- 1 standard deviation instead of just + 1 std.

Table S2: Column 2 you use MNB instead of NMB

---

## Referee Comment (RC2) · Anonymous Referee #2 · 18 Sep 2018

In this paper, the authors describe updates made to aerosol representations in the Oslo Chemistry Transport Model, compare the simulated aerosol to various observations, and use that model with a recent emission dataset to derive the time series of radiative forcing of aerosol-radiation interactions (RFari) over the industrial era. Their estimate of RFari is at the weaker end of latest AeroCom and IPCC assessments, and the authors work suggest that globally-averaged RFari has weakened since the 1980s.

The paper is nicely written and to the point. Figures are well chosen and illustrate the discussion well. Sensitivity studies are interesting. There are a few aspects that can

be clarified however: The justification for model parameters is often lacking and there is also a need to better link section 3.3 on RFari timeseries with model evaluation.

Below I list the clarifications that I would like to see ahead of the paper's publication. Addressing those comments should not require additional analysis, so represent minor revisions.

**1   Comments**

- Abstract, line 38: Be more specific about what is new about the treatment of black carbon in OsloCTM3

- Abstract, line 47: Aren't the AR5 forcing estimates for 2011?

- Introduction, line 76: Are those the CMIP5 historical emissions documented by Lamarque et al., doi:10.5194/acp-10-7017-2010, 2010? If so, then reference the paper here.

- Introduction, line 77: I seem to remember that historical finished in 2005, not 2000, in CMIP5.

- Introduction, lines 114–116: Is that sentence still talking about scavenging?

- Section 2.1, line 157: What are the finer spatial and temporal resolutions of the new transfer rates?

- Section 2.1, lines 161–163, line 203, and Table 2; and Section 2.4, lines 284–285: The numbers selected for OC-OM conversion, hydrophilic fractions of BC and OM, aerosol solubility fractions, and absorbing OM fractions need to be justified. Where do they come from?

- Section 2.1, lines 169–170: How is the scaling of marine OM emissions done in practice? Sea spray is dependent on wind speed, so do you need to run the model a first time to get OM emissions, then scale them, then run again?

- Section 2.1, line 195: Why does the DEAD scheme need energy budget calculations?

- Section 2.1, paragraph starting line 198: Does the model account for re-evaporation of precipitation? And re-suspension of dry-deposited aerosol?

- Section 2.3, line 249: Why fix the meteorology to 2010? Once reanalysis data is available (from 1979 for ERA, I believe), then it would be good to use the meteorology that corresponds to the actual year. There is a sensitivity study dedicated to the impact of meteorology, and that impact is fairly sizeable (lines 556–558). So using 2010 for all years does not seem ideal.

- Section 2.3, paragraph starting line 254: It would be good to have more insight into the choice of sensitivity studies of removal. Why no SOLINC sensitivity study, for example?

- Section 2.3, line 258: I do not think that the resolution of the control simulation has been given at this stage, but I may have missed it.

- Section 2.4, paragraph starting line 270: It would be useful to have a Table summarising the assumptions made about size distribution and refractive index (or MEC/MAC if that is what the model uses) for all modelled species, with references where appropriate. That information is crucial to understanding differences in radiative forcing efficiencies between models, yet is rarely given in model description papers.

- Section 2.5, line 297: "*frameworks*" – did you mean networks?

- Section 2.5, lines 321–322: What does the AeroCom validation tool do?

- Section 3.1 starting line 354: It would be most useful to also include residence times for each species in Table 3. Perhaps in parentheses after the burden? Residence times are also crucial to understanding differences in aerosol distributions among models.

- Section 3.1, paragraph starting line 356: The authors try to explain differences with the previous version of the model, which is great, but that is not done consistently. Why is sulfate burden 35% higher than in OsloCTM2? (Line 360.) Is that understood? And what explains the 25% higher OA burden? According to lines 369–370, the marine POA contribution is too small to explain that large difference.

- Section 3.1, line 377: "*against output*" – it is the other way around, model versus observations.

- Section 3.1, line 384: What correlation? Need to rephrase to clarify that sentence.

- Section 3.1, line 414: Is being close to the AeroCom nitrate multi-model mean a good thing? How do models compare to observations in Bian et al. 2017?

- Section 3.1, lines 426–428: Erroneous emissions have a big impact on model performance. Can't you re-run at all? That would be a shame.

- Section 3.1, Figure 2: Could you show numbers in each slice of the pie chart in Figure 2? That would help make the comparison more quantitative.

- Section 3.1, paragraph starting line 453: Figure 3d suggests that the model overestimates low optical depths but underestimates large optical depths. Is that correct? From experience, it is a common model deficiency, It may have even been noted in an AeroCom paper.

- Section 3.1, line 468: Small addition for clarity: "*there are notable difference* in model performance"

- Section 3.2, line 502: The analysis should point to Table 2 to make sense of sensitivities to wet removal. Species with the same scavenging fractions are likely to behave similarly in such sensitivity studies.

- Section 3.2, lines 519–520 and Figure 4: What is happening in North America? The change is in the Midwest, whereas I would have put the source regions more to the East.

- Section 3.2, lines 526–527 and Figure 4: What is happening with dust in MET-DTA? Also a teleconnection with ENSO?

- Section 3.2, lines 530–531: Why such a strong dependence of nitrate on emission inventory? The vertical profile of ammonium nitrate formation is determined by temperature and competition from ammonium sulfate, so the role of emission inventories is not obvious. Which emission differences matter: ammonia or sulfur dioxide?

- Section 3.2, lines 537–540: Again, should link to Table 2.

- Section 3.3 starting line 618: It would most interesting to link specific model deficiencies to errors in RFari estimates. This is done briefly in the discussion, but could be done more clearly in section 3.3. For example, he model generally underestimate surface concentrations. Does that bias cancel out when taking differences over the industrial era? Another example is that the model underestimates aerosol in Asia. Does that mean that recent RFari is underestimated? Or that similar underestimations would have happened over US/Europe when sulfate was high in those regions in the 1950-1980s? A final example is about high latitudes. There is a confident conclusion about RFari changes north of 70°N (lines 679–681), but do you really have confidence in your model in that region?

- Section 3.3, line 638: What is "*OsloCTM3fast*"?

- Section 3.3, line 675: "*clearly*" – Not that strong a signal really. Did the radiative forcing efficiency change with the change in emitting region? Combining by eye the two panels of Figure 8 would suggest that the change is rather small.

-

---

## Author Comment (AC1) · 11 Oct 2018

Response to review of *"Concentrations and radiative forcing of anthropogenic aerosols from 1750-2014 simulated with the OsloCTM3 and CEDS emission inventory"* by Marianne T. Lund, Gunnar Myhre, Amund S. Haslerud, Ragnhild B. Skeie, Jan Griesfeller, Stephen M. Platt, Rajesh Kumar, Cathrine Lund Myhre, Michael Schulz

We thank the anonymous referee for the careful and thorough review of our paper, and the useful suggestions. Responses to individual comments are given below.

**Anonymous Referee #2**

In this paper, the authors describe updates made to aerosol representations in the Oslo Chemistry Transport Model, compare the simulated aerosol to various observations, and use that model with a recent emission dataset to derive the time series of radiative forcing of aerosol-radiation interactions (RFari) over the industrial era. Their estimate of RFari is at the weaker end of latest AeroCom and IPCC assessments, and the authors work suggest that globally-averaged RFari has weakened since the 1980s. The paper is nicely written and to the point. Figures are well chosen and illustrate the discussion well. Sensitivity studies are interesting. There are a few aspects that can be clarified however: The justification for model parameters is often lacking and there is also a need to better link section 3.3 on RFari timeseries with model evaluation. Below I list the clarifications that I would like to see ahead of the paper's publication. Addressing those comments should not require additional analysis, so represent minor revisions.

**1 Comments**
• Abstract, line 38: Be more specific about what is new about the treatment of black carbon in OsloCTM3
Clarified. The text now reads:
*"The treatment of black carbon (BC) scavenging in OsloCTM3"*

• Abstract, line 47: Aren't the AR5 forcing estimates for 2011?
Yes, this is a typo, thanks for noticing.

• Introduction, line 76: Are those the CMIP5 historical emissions documented by Lamarque et al., doi:10.5194/acp-10-7017-2010, 2010? If so, then reference the paper here.
Yes, reference included.

• Introduction, line 77: I seem to remember that historical finished in 2005, not 2000, in CMIP5.
Corrected. In fact, this sentence confused the inventory documented by Lamarque (1850-2000) with the CMIP5 experimental design with historical period 1850-2005. For clarification, we have rephrased slightly, adding referring to both Lamarque et al. 2010 and Taylor et al. 2012. The paragraph now reads:

*"The Community Emissions Data System (CEDS) recently published a new time series of emissions from 1750 to 2014, which will be used in the upcoming CMIP6 [Hoesly et al., 2018]. CEDS includes several improvements, including annual temporal resolution with seasonal cycles, consistent methodology between different species, and extending the time series to more recent years, compared to previous inventories and assessments [e.g., Lamarque et al., 2010; Taylor et al., 2012]."*

• Introduction, lines 114–116: Is that sentence still talking about scavenging?

Scavenging and other processes. Text clarified:
*"(…) few studies have focused on impacts of scavenging and other processes on a broader set of aerosol species or the combined impact in terms of total aerosol optical depth (AOD)."*

• Section 2.1, line 157: What are the finer spatial and temporal resolutions of the new transfer rates?
Monthly instead of seasonal and latitudinal rates established based on simulations of 10 instead of four emission source regions, covering not only the northern hemisphere, but giving near global coverage. Now specified in the text:
*"Specifically, the latitudinal transfer rates have been established based on experiments with 10 instead of four emission source regions and with monthly, not seasonal resolution."*

• Section 2.1, lines 161–163, line 203, and Table 2; and Section 2.4, lines 284–285: The numbers selected for OC-OM conversion, hydrophilic fractions of BC and OM, aerosol solubility fractions, and absorbing OM fractions need to be justified. Where do they come from?
References to OC-OM conversion and hydrophilic fractions of BC and OM added.

On the OM fraction we have added the following:
"*Organic matter has a large variation in the degree of absorption [e.g., Kirchstetter et al., 2004; Xie et al., 2017], from almost no absorption to a strong absorption in the ultraviolet region. Here, we have implemented absorbing organic matter according to refractive indices from Kirchstetter et al. [2004]. The degree of absorption varies by source and region and is at present inadequate quantified: Here we assume 1/3 of the biofuel organic matter and ½ of the SOA from anthropogenic volatile organic carbon (VOC) precursors. The remaining fractions of biofuel, fossil fuel and marine POA and SOA (anthropogenic and all natural VOCs) are assumed to be purely scattering organic matter. As these fractions are not sufficiently constrained by observational data and associated with significant uncertainty, we also perform calculations with no absorption by OA."*

• Section 2.1, lines 169–170: How is the scaling of marine OM emissions done in practice? Sea spray is dependent on wind speed, so do you need to run the model a first time to get OM emissions, then scale them, then run again?
The scaling is done by multiplying with a fixed factor, determined from previous test runs, which depends on the sea salt production scheme used (here the factor is 0.5). The exact scaling depends on meteorology and resolution, found to result in inaccuracies of 3-5%. The marine OM emissions is checked after the run to make sure that no additional tuning of the scaling factor is needed beyond these uncertainties. We have specified the scaling factor in the text based on comments by both referees. The following has been added:
*"The scaling factor depends on the chosen sea salt production scheme (see below) and to some degree on the resolution; here we have used a factor of 0.5. "*

• Section 2.1, line 195: Why does the DEAD scheme need energy budget calculations?
Radiative fluxes are needed for calculation of boundary layer properties related to the dust mobilization. For clarification we have added:
*"As a minor update, radiative flux calculations, required for determination of boundary layer properties in the dust mobilization parameterization [Zender et al., 2003], now uses radiative surface properties and soil moisture from the meteorological fields."*

• Section 2.1, paragraph starting line 198: Does the model account for reevaporation of precipitation? And re-suspension of dry-deposited aerosol?

The model treats evaporation (reversible and irreversible), but not re-suspension. Following this comment, and comment from anonymous referee #1, we have expanded the description of removal in Sect. 2.1, including these specifications.

• Section 2.3, line 249: Why fix the meteorology to 2010? Once reanalysis data is available (from 1979 for ERA, I believe), then it would be good to use the meteorology that corresponds to the actual year. There is a sensitivity study dedicated to the impact of meteorology, and that impact is fairly sizeable (lines 556–558). So using 2010 for all years does not seem ideal.

Because re-analysis data is only available for a limited period of the historical time series considered here, we maintain that keeping meteorological data constant is a reasonable experimental design for a study with our objective, for consistency and in order to isolate the effect of change in the emissions. Moreover, RF is calculated relative to pre-industrial, where we have no way of using "real" meteorological data. As recent study by Myhre et al. 2017 looked at the effect of emission changes over the 1990-2014 period, and in this case meteorology for each year was used in some of the models.

Our sensitivity test gives an indication of the order of magnitude impact on the aerosol burden and, as the direct RF scales quite linearly with aerosol burden, on the consequent RF. Using the rather extreme case of opposite ENSO phases, we find burden changes of 1-10%, suggesting that using constant meteorological input data is unlikely to change conclusions about the overall trends

Use of correct meteorological data is of course important for accurate distribution of concentrations and model evaluation, which is why we limit our validation to the year 2010.

• Section 2.3, paragraph starting line 254: It would be good to have more insight into the choice of sensitivity studies of removal. Why no SOLINC sensitivity study, for example?

The following text has been added for clarification:

*"To modify the scavenging, we tune the fixed fractions that control aerosol removal efficiency in the model (see Sect. 2.1). Table 2 summarizes fractions used in the baseline configuration and the three sensitivity tests. A decrease and increase in efficiency of 0.2 is adopted for scavenging of all aerosols by liquid clouds (except hydrophobic BC and OM) and ice clouds, respectively. Note that there is no test with increased removal by liquid clouds, as, with the exception of hydrophobic BC, OM and SOA, 100% efficiency is already assumed. For ice clouds, we also reduce the efficiency to a fraction of 0.1, or 0.001 if the value is 0.1 in the baseline configuration. We note that these changes do not represent realistic uncertainty ranges based on experimental or observational evidence, as there are limited constraints in the literature, but are chosen to explore the impact of a spread in the efficiency with which aerosols act as ice and cloud condensation nuclei."*

• Section 2.3, line 258: I do not think that the resolution of the control simulation has been given at this stage, but I may have missed it.

The resolution is given in the paragraph before.

• Section 2.4, paragraph starting line 270: It would be useful to have a Table summarizing the assumptions made about size distribution and refractive index (or MEC/MAC if that is what the model uses) for all modelled species, with references where appropriate. That information is crucial to understanding differences in radiative forcing efficiencies between models, yet is rarely given in model description papers.

We believe that this information is readily enough available from the references (Myhre et al. 2007, Myhre et al. 2009) in Sect. 2.4, with updates subsequently described. A table with this information has been included in the supplementary material. However, we have added a table in the supplementary material with changes in burden, AOD, AAOD, RFari, and normalized RF over the period 1750-2010 given for individual aerosol components, as well as the net RFari. This information has been commonly presented in previous studies and will allow the reader to better understand differences compared to the present analysis. The text has been updated accordingly.

• Section 2.5, line 297: "frameworks" – did you mean networks?
Yes, error corrected.

• Section 2.5, lines 321–322: What does the AeroCom validation tool do?
Probably poor/unclear chose of wording: The comparison between the model and measurement is done through the AeroCom data base, i.e., model data is interpolated and extracted for each location of the AERONET stations (this method is already described in a later paragraph). We have clarified the sentence:
*"The comparison with AERONET data was done using processed through the validation tools available from the AeroCom data base hosted by Met Norway."*

• Section 3.1 starting line 354: It would be most useful to also include residence times for each species in Table 3. Perhaps in parentheses after the burden? Residence times are also crucial to understanding differences in aerosol distributions among models.
Good suggestion, we agree that this is useful additional information. We have included the residence times for the baseline simulation in Table 3, with remaining simulations in a separate table in the supplementary material (for readability).

• Section 3.1, paragraph starting line 356: The authors try to explain differences with the previous version of the model, which is great, but that is not done consistently. Why is sulfate burden 35% higher than in OsloCTM2? (Line 360.) Is that understood? And what explains the 25% higher OA burden? According to lines 369–370, the marine POA contribution is too small to explain that large difference.
We have added the following for sulfate:
*"While the total $SO_2$ emission is only 5% higher in the present study than in the OsloCTM2 AeroCom III simulation, the atmospheric residence time of sulfate is 50% longer, suggesting that the burden difference is mainly attributable to changes in the parameterization of dry and large-scale wet deposition (Sect. 2.1)."*
And modified the OA discussion:
*"The OsloCTM3 estimate includes the contribution from marine OA emissions (Sect. 2.1), which may explain part of the difference as marine OA was included in some of the AeroCom II models, but not OsloCTM2. Additionally, the residence time of OA of 5.3 days is longer than in the OsloCTM2 AeroCom II experiment."*

• Section 3.1, line 377: "against output" – it is the other way around, model versus observations.
The sentence is technically correct as it stands, I think, but I can see that it would be more as expected to start with the model output and have switched it around for clarification:
*"Figure 1 shows results from the baseline OsloCTM3 simulation against annual mean measured surface concentrations of EC, OC, sulfate and nitrate in Europe, North America and Asia."*

• Section 3.1, line 384: What correlation? Need to rephrase to clarify that sentence.
With CAWNET observations. Rephrased for clarification:
*"In contrast, the correlation with CAWNET observations is generally similar to, or higher than, other regions/networks."*

• Section 3.1, line 414: Is being close to the AeroCom nitrate multi-model mean a good thing? How do models compare to observations in Bian et al. 2017?
Good point. We have added the following:
*"Results showed that  most models tend to underestimate ammonium concentrations compared to observations in North America, Europe and East Asia, with a multi-model mean bias and correlation of*

*0.886 and 0.47, respectively. Tthe OsloCTM3 shows good agreement with ammonium measurements in North America, but has a bias and correlation close the model average in the other two regions."*

• Section 3.1, lines 426–428: Erroneous emissions have a big impact on model performance. Can't you re-run at all? That would be a shame.

We did re-run the year 2010 simulation for the assessment of impact described in this paragraph. In the revised manuscript, the documentation of the impact of the updated CEDS emissions on model performance is also extended to more species and the following text has been added:

*"While repeating all simulations require more resources than available, we have performed an additional run for the year 2010. Figure S4 shows the comparison of modeled concentrations against IMPROVE measurements with the two emission inventory versions, CEDSv16 and CEDSv17. In the case of BC, the comparison shows a 5% higher correlation and 15% lower RMSE with the CEDSv17 than CEDSv16. A similar improvement is found for nitrate, with 26% higher correlation and 12% lower RMSE, while in the case of OC and sulfate, the difference is small (< 5%). Smaller differences of between 2-10% are also found in the comparison against measurements in Europe and Asia (not shown). Hence, using the updated version of the emission inventory has an effect on the model performance in terms of surface concentrations, but without changing the overall features or conclusions."*

As global and national emission totals are unchanged between the two inventory versions, and we consider only the direct radiative effect, we believe that the implications for the RF estimates is small. We confirm this by also repeating the 1750 simulation as part of the revision: the RFari (2010-1750) is 2% stronger with the CEDSv17 inventory.

• Section 3.1, Figure 2: Could you show numbers in each slice of the pie chart in Figure 2? That would help make the comparison more quantitative.

Yes, done.

• Section 3.1, paragraph starting line 453: Figure 3d suggests that the model overestimates low optical depths but underestimates large optical depths. Is that correct? From experience, it is a common model deficiency, It may have even been noted in an AeroCom paper.

Yes, the model overestimates the very lowest measured AOD values. The most pronounced underestimation is seen for AOD around 0.1 to 1.0, but with a tendency towards more underestimation than overestimation also for the highest measured values. However, we are not aware of any AeroCom papers or multi-model studies documenting similar results that we could base a discussion around.

• Section 3.1, line 468: Small addition for clarity: "there are notable difference in model performance"

Included.

• Section 3.2, line 502: The analysis should point to Table 2 to make sense of sensitivities to wet removal. Species with the same scavenging fractions are likely to behave similarly in such sensitivity studies.

Good point. Included.

• Section 3.2, lines 519–520 and Figure 4: What is happening in North America? The change is in the Midwest, whereas I would have put the source regions more to the East.

Upon closer inspection of the burden of individual components, we see that this increase in total AOD in Central North America is mainly driven by an increase in ammonium nitrate. There is actually a dipole pattern in the ammonium nitrate change with different emission inventories: a lower burden is found in the Easternmost US (i.e., east of approx. 90W) and a notable increase in burden in the Midwest when ECLv5 and CMIP5 emissions are used. The former is likely due to the higher SO2 emissions in these inventories than in CMIP6, which is mainly confined to the east and south along the Mexican Gulf. The increased burden likely results from the increase in NOx emissions, which extends into the states of

Kansas, Oklahoma and Nebraska, and beyond. At the same time there is slight decrease in SO2 emissions in the western part of the US, which reduces sulfate concentration and hence competition for available ammonia. In the ECLv5 inventory, there are also higher emissions of NH3. We have added the following to the discussion:

*"Over central North America the AOD is higher, mainly due to more ammonium nitrate, whereas the higher AOD over Eastern Europe and part of Russia is a result on higher sulfate concentrations."*

• Section 3.2, lines 526–527 and Figure 4: What is happening with dust in METDTA? Also a teleconnection with ENSO?
Presumably, this refers to the ratio of total AOD in figure 4 and the area with values above one off the coast of North Africa, where the dust influence on AOD is prominent. The influence of met-data on dust occur through both production and transport/scavenging. There are some studies suggesting impacts of ENSO on precipitation in Sahel, and given that we have selected years of opposite ENSO phase, it is likely that we see an impact. However, other meteorological differences will also play a role and cannot be distinguished. While we cannot provide a quantitative answer, we have added the following to the discussion:

*"There is also a notable change in the Atlantic Ocean, where mineral dust is a dominating species. The meteorological data can affect production, deposition and transport of dust directly, as well as indirectly through ENSO-induced teleconnections as suggested by e.g., Parhi et al. [2016]."*

• Section 3.2, lines 530–531: Why such a strong dependence of nitrate on emission inventory? The vertical profile of ammonium nitrate formation is determined by temperature and competition from ammonium sulfate, so the role of emission inventories is not obvious. Which emission differences matter: ammonia or sulfur dioxide?
The change in nitrate formation is a complex interplay of changes in several gases (SO2, NH3 and NOx) and the differences in these, in turn, vary between inventory, as well as region. Hence, a significant impact of emission inventory is not unexpected. However, it is interesting to note the stronger effect on emissions, and less impact of changes to the scavenging, on the burden of nitrate, and the contrast to the other species. Based on this comment and the one above, we have added the following:

*"In general, the burden of BC, OA and dust is significantly affected by changes in the scavenging assumptions, while nitrate responds more strongly to different emission inventories, likely due to the complicated dependence on emissions of several precursors and competition with ammonium-sulfate. We also note that at higher altitudes the absolute differences in the burden of nitrate are small."*

With the data available from the simulations in the present study, we cannot quantify the roles of the respective emission difference. There are also likely strong seasonal and geographical variations. However, an inspection of the spatial distribution of differences in emissions, zonal mean concentrations and burden, can shed some light on the driver. It also important to point out that at high altitudes, nitrate values are small, as are the absolute differences.

For RCP/CMIP5 emissions, the global mean vertical profile of nitrate is lower than then using CMIP6 emissions throughout the atmosphere, mainly determined by a decrease in concentration between 10 and 40N. Compared to the CMIP6 inventory, RCP/CMIP5 emissions of both NH3 are lower in Asia, North Africa and South America. NOx is also lower, in particular in Asia. (SO2 emissions in India are somewhat lower as well, but we do not know to what extent less completion for available ammonia offsets an overall lower amount of precursors.) NOx is also lower in South Asia in ECLv5, while sulfate is higher, the combined effect being a lower nitrate burden. However, for nitrate there is a higher concentration around 15-40S and 35-50N, especially 800 and 400 hPa. This seems to arise, at least in part, from higher NH3 emissions, and areas with lower SO2, in China, South America and central Africa. So in summary, changes to all three gases seem to contribute, but with different roles in different regions.

• Section 3.2, lines 537–540: Again, should link to Table 2.
Done.

• Section 3.3 starting line 618: It would most interesting to link specific model deficiencies to errors in RFari estimates. This is done briefly in the discussion, but could be done more clearly in section 3.3. For example, he model generally underestimate surface concentrations. Does that bias cancel out when taking differences over the industrial era? Another example is that the model underestimates aerosol in Asia. Does that mean that recent RFari is underestimated? Or that similar underestimations would have happened over US/Europe when sulfate was high in those regions in the 1950-1980s? A final example is about high latitudes. There is a confident conclusion about RFari changes north of 70_N (lines 679–681), but do you really have confidence in your model in that region?
It is not straightforward to investigate indications of biases from comparisons to observations and the implication for RF. A bias compared to surface measurements may not necessarily imply that the total column has the same bias. Moreover, even if there is an underestimation of the total AOD, the influence of such a bias on RF depends on the mix of absorbing and scattering aerosol. Similarly, the aerosol mix is an important complicating factor the effect of underestimated or missing emission sources, for instance in Asia, where the uncertainties in the emissions are larger than in other regions. At high latitudes, the model does show an underestimation of BC concentrations, both at the surface and at higher altitudes. However, this is mainly an issue during winter and early spring, when the direct aerosol effect is small due to lack of sunlight. For other species and AOD, the availability of observations is a limiting factor. Additionally, we cannot perform a detailed evaluation of the simulated pre-industrial concentrations.

As the referee notes, the manuscript already discusses some of these issues. For the reasons noted above, we are reluctant to speculate too much beyond these more general points or attempt too specific links between model deficiencies and RF estimates. However, we have restructured Section 4 slightly, to make this discussion easier to follow and included a couple of additional issues. The paragraph now reads:

*"A significant range from -0.6 to -0.13 W m-2 surrounds the central RFari estimate of -0.35 W m-2 from IPCC AR5 [Boucher et al., 2013], caused by the large spread in underlying simulated aerosol distributions. Deficiencies in the ability of global models to reproduce atmospheric aerosol concentrations can propagate to uncertainties in RF estimates. As shown in Sect. 3, the OsloCTM3 generally lies close to or above the multi-model mean of anthropogenic aerosol burdens from recent studies and is found to perform reasonably well compared with observations and other global models, with improvements over the predecessor OsloCTM2. In particular, recent progress towards constraining the vertical distribution of BC concentrations has resulted in improved agreement between modeled and observed vertical BC profiles over the Pacific Ocean with less of the high-altitude overestimation seen in earlier studies. However, as shown by Lund et al. [2018], there are discrepancies compared to recent aircraft measurements over the Atlantic Ocean. A remaining challenge is the model underestimation of Arctic BC concentrations. However, this is seen mainly during winter and early spring, when the direct aerosol effect is small due to lack of sunlight. In contrast, the higher emissions in the CEDSv16 inventory also results in an improved agreement with BC surface concentrations over Asia. In general, we find lower surface sulfate concentrations in the model compared with measurements. This could contribute to an underestimation of the sulfate RFari, which is weaker in the present study than in IPCC AR5. An underestimation of observed AOD in Asia is also found, however, the implication of this bias on RF is not straightforward to assess, as it is complicated by the mix of absorbing and scattering aerosols. (…)"*

• Section 3.3, line 638: What is "OsloCTM3fast"?
This is a typo, there is no fast version of the model. Corrected.

• Section 3.3, line 675: "clearly" – Not that strong a signal really. Did the radiative forcing efficiency change with the change in emitting region? Combining by eye the two panels of Figure 8 would suggest that the change is rather small.

We think that the decline in net RFari north of 40°N is quite clearly seen in this figure, as is the strengthening between 10-30°N. However, the peak RFari is indeed significantly lower in 2014 than it was in 1970-90s. This is in large part due to the different evolution of scattering and absorbing aerosols, where sulfate burden is lower than in the late 20$^{th}$ century, while the BC burden has contributed to increase. There may also be an influence of different forcing efficiencies, however, since we only consider direct radiative effects, it is likely to be smaller than the impact of changing relative importance of scattering and absorbing aerosols. To clarify, we have modified the text:

*"Over the past decades, there has been shift in emissions, from North America and Europe to South and East Asia. This is also reflected in the zonally averaged net RFari over time in Fig. 8b. RFari declined in magnitude north of 40°N after 1980, with particularly large year-to-year decreases between 1990 and 1995, and from 2005 to 2010, and strengthened in magnitude between 10°-30°N. The RFari also strengthened in the Southern Hemisphere subtropical region, reflecting incresing emission in Africa and South America after 1970. However, the peak net RFari is considerably weaker in 2014 than the peak in 1980. This mainly is due to fact that simultaneously with the southwards shift, the sulfate burden has declined, while the BC burden has increased steadily at the same latitudes, resulting in a weaker net forcing."*

---

## Author Comment (AC2) · 11 Oct 2018

Response to review of *"Concentrations and radiative forcing of anthropogenic aerosols from 1750-2014 simulated with the OsloCTM3 and CEDS emission inventory"* by Marianne T. Lund, Gunnar Myhre, Amund S. Haslerud, Ragnhild B. Skeie, Jan Griesfeller, Stephen M. Platt, Rajesh Kumar, Cathrine Lund Myhre, Michael Schulz

We thank the anonymous referee for the careful and thorough review of our paper, and the useful suggestions. Responses to individual comments are given below.

**Anonymous Referee #1**

**General comments**:
This paper evaluates the present-day aerosol distributions and radiative forcing in the latest OsloCTM3 chemical transport model. Surface concentrations, vertical distributions as well as aerosol optical depth are evaluated for the year 2010 against an extensive set of ground-based and satellite measurements. Uncertainties in the aerosol distributions are also assessed through a number of sensitivity studies assessing the uncertainty in the aerosol emissions, removal processes, meteorology and model resolution. The model is then used to assess the historical aerosol direct radiative forcing (RF) from 1750-2010. The net radiative forcing as well as the RF due to individual species are compared against the most recent published literature and CMIP5 estimates. Aerosol-cloud interactions are not assessed in this study.

This is a well-written, clearly presented, model evaluation manuscript. This paper does not document the latest updates to the OsloCTM3 model, but appropriate references are provided for this. Instead it documents the latest aerosol simulations with this model using the latest CMIP6 emissions inventory and subsequently derives the latest direct aerosol RF estimate. So I believe it is within the GMD scope as a Model Evaluation paper. This paper is timely given the CMIP6 project is now well underway and will provide a useful update to direct aerosol RF estimates in this regard. I recommend this paper to be published subject to a number of revisions I detail below.

**Specific comments**:
I struggled to appreciate the motivation for the additional sensitivity simulations conducted in this study. While these studies are useful and worth reporting the motivation for conducting these studies and link with the rest of the paper needs to be made clearer. This is probably most easily achieved in the Introduction.
Parts of the introduction has been rewritten in order to address this in a better and more explicit way. In particular, the following paragraph has been added:

*"As accurate representation of the observed aerosol distributions in global models is crucial for confidence in estimates of radiative forcing (RF), these issues emphasize the need for broad and up-to-date evaluation of model performance.*
*The diversity of simulated aerosol distributions, and discrepancies between models and measurements, stem from uncertainties in the model representation aerosol processing. Knowledge of the factors that control the atmospheric distributions is therefore needed to identify potential model improvements and need for further observational data, and to assess how remaining uncertainties affect the modeled aerosol abundances and, in turn, estimates of RF and climate impact. A number of recent studies have investigated the impact of changes in aging and scavenging processes on BC distribution, focusing on aging and wet scavenging processes (e.g., [Bourgeois and Bey, 2011; Browse et al., 2012; Fan et al.,*

*2012; Hodnebrog et al., 2014; Kipling et al., 2013; Lund et al., 2017; Mahmood et al., 2016]), resulting in notable improvements, at least for specific regions or observational data sets. However, with some notable exceptions (e.g., [e.g., Kipling et al., 2016]), fewer studies have focused on impacts of scavenging and other processes on a broader set of aerosol species or the combined impact in terms of total aerosol optical depth (AOD)."*

In Sect 2.3 Please report the length of each time slice simulation and how this impacts the signal to noise in resulting RF estimates. Are you running 20-30 years for each time slice?
Each simulation is one year, with six months spin up. Text has been clarified and now reads:
*"All simulations are one year with six months spin-up. "*

In Sect 3.3 please describe clearly either explicitly or through appropriate references how you calculated the RFari. Have you calculated an effective radiative forcing or used the more traditional radiative forcing metric? It is currently unclear. Also, how are the individual species RFari determined - through species only runs or is the RT code able to output this?. Please provide this detail in the manuscript.
Regarding the first half of this comment: The radiative forcing calculations described in detail in Sect. 2.4 of the paper, including a definition of RFari, i.e., forcing due to aerosol-radiation interactions, not including any rapid adjustments as for the effective RF. To clarify, we refer to back to the description in Sect. 3.3. The species-specific RFari is obtained by individual runs, where the concentration of the respective species is set to the pre-industrial level. This has been clarified in the text:
*"The RFari of individual aerosols is obtained by separate simulations, where the concentration of the respective species is set to the pre-industrial level."*

Use of older version of CEDS emission inventory: I am not convinced that the historical RFari will not be impacted at all by the choice of CEDS emission inventory. Changes in spatial distribution of emission could impact aerosol removal, transport and thus lifetime, temporal shifts in the distribution could also potentially impact the historical evolution of the ERF. While I understand the computational burden of repeating all tests, two runs using the new CEDS version with 1750 and for instance year 2010 emissions would allow you to quantify the impact on the RF fairly easily to allow you to justify it in the text. It looks like you may have the 2010 simulation already from you Fig S4 plots.
We have repeated the 1750 simulation with the new CEDS version 2017 inventory and calculate a 2% stronger net RFari (2010 relative to 1750) than with the 2016 version. Global burdens differ by 2-10% between runs with the two inventories, showing that there is some impact through removal and transport. The impact on net RFari is a combination of a slightly higher BC burden and lower sulfate and OA burden. Here limit the analysis to direct aerosol effects only. RFari scales more with the total burden changes, which are small. As the reviewer points out, the effect of spatial changes in the aerosol distribution could be more important for the ERF, i.e., also considering aerosol-cloud interactions.

Furthermore limiting your evaluation to BC is also questionable as I would expect notable differences in for instance SO2. I would request that you extend this evaluation over the USA to all aerosol species where you have observations.
We agree and have extended the comparison to organic aerosol, nitrate and sulfate, as well measurements over in Asia and Europe. Figure S4 has been updated and the following text is added:
*"While repeating all simulations require more resources than available, we have performed an additional run for the year 2010. Figure S4 shows the comparison of modeled concentrations against IMPROVE measurements with the two emission inventory versions, CEDSv16 and CEDSv17. In the case of BC, the comparison shows a 5% higher correlation and 15% lower RMSE with the CEDSv17 than CEDSv16. A similar improvement is found for nitrate, with 26% higher correlation and 12% lower RMSE, while in the case of OC and sulfate, the difference is small (< 5%). Smaller differences of between 2-10% are also found*

*in the comparison against measurements in Europe and Asia (not shown). Hence, using the updated version of the emission inventory has an effect on the model performance in terms of surface concentrations, but without changing the overall features or conclusions. The net RFari in 2010 relative to 1750 is 2% stronger with the CEDSv17 inventory, a combined effect of slightly higher global BC burden and lower burdens of sulfate and OA."*

Changes to the large-scale ice scavenging efficiency have a large control on the global aerosol distribution. The paper would benefit from a more detailed description of how the LS ice (and liquid) scavenging is parameterized in the model in Sect 2.1
The paragraph on removal has been expanded with more details and now reads:

*"Aerosol removal includes dry deposition and washout by convective and large-scale rain. Rainfall is calculated based on European Center for Medium-Range Weather Forecast (ECMWF) data for convective activity, cloud fraction and rain fall. The efficiency with which aerosols are scavenged by the precipitation in a grid box is determined by a fixed fraction representing the fraction of this box that is available for removal, while the rest is assumed to be hydrophobic. The parameterization distinguishes between large-scale precipitation in the ice and liquid phase, and the OsloCTM3 has a more complex cloud model than OsloCTM2 that accounts for overlapping clouds and rain based on [Neu and Prather, 2012]. When rain containing species falls into a grid box with drier air it will experience reversible evaporation. Ice scavenging, on the other hand, can be either reversible or irreversible. For further details about large-scale removal we refer the reader to Neu and Prather [2012]. Convective scavenging is based on the Tiedtke mass flux scheme (Tiedtke 1989) and is unchanged from the OsloCTM2. The solubility of aerosols is given by constant fractions, given for each species and type of precipitation (i.e., large-scale rain, large-scale ice, and convective) (Table 2). Dry deposition rates are unchanged from OsloCTM2, but the OsloCTM3 includes a more detailed land use dataset (18 land surface categories at 1°x1° horizontal resolution compared to 5 categories at T42 resolution), which affects the weighting of deposition rates for different vegetation categories. Re-suspension of dry deposited aerosols is not treated."*

Technical corrections:
Line 70: 2011 - this should be 2010
IPCC AR5 give the forcing in 2011 relative to 1750. However, we see that there is a typo in our abstract, where 2010 is used. This has been corrected.

Line 170: It would be worthwhile to report here what global scaling factor you have used in the Gantt parametersation of marine OM. Gantt et al. 2015 I think use a global scaling factor of 6 but this is believed to be highly model dependent.
This is good point. Using the scaling factor of 6 from Gantt et al. produces too high emissions in our model. The scaling factor is also dependent on the sea salt production scheme and, to some degree, on resolution. With the sea salt scheme used here, we use a factor of 0.5. The text has been updated and reads:

*"The scaling factor depends on the chosen sea salt production scheme (see below) and to some degree on the resolution; here we have used a factor of 0.5. "*

Line 281: Explicity state the aerosol concentration threshold below which you apply the Bond and Bergstrom method. Why do you change approaches? Again state the motivation for these different approaches in different regimes and why they need to be made. Does Zanatta lead to too high a MAC in

the low aerosol regimes? Has this been constrained by observations - if yes, provide appropriate reference here. Some discussion of the uncertainties in both approaches as this links to your uncertainties in BC RFari discussed in Sect 3.3

We have included the following description in the manuscript:

*"The measurements in Zanatta et al. [2016] represent continental European levels. For very low concentrations of BC, the formula given in Zanatta et al. [2016] provides very high MAC values. We have therefore set a minimum level of BC of 1.0e-10 g m$^{-3}$ for using this parameterization, and for lower concentrations we use [Bond and Bergstrom, 2006]. In addition, we have set a maximum value of MAC of 15 m$^2$ g$^{-1}$ (637 nm) to avoid unrealistic high values of MAC compared to observed values."*

Line 329-330: It would be useful here to be clear on what model variables you've sampled at 3 hour resolution and what ones you've sampled monthly and why you needed to do this.

Text has been clarified:

*"the model data is linearly interpolated to the location of each station using annual mean, monthly mean (concentrations) or 3-hourly output (AOD), depending on the resolution of the observations."*

Line 452: EBAS - I think this should refer to EMEP/ACTRIS. EBAS has not been previously mentioned.

Yes, thank you for pointing this out.

Line 468: Remove "as for surface concentrations,"

Removed.

Table 3: It would be useful to have the % change in burden listed alongside the burdens of all the sensitivity experiments. Perhaps in parentheses next to the burdens.

We tried this, but we think that it exacerbated the readability of the table without adding much useful information as these numbers are easily calculated from the absolute values. We have therefore chosen to leave this table as is.

Line 541-544: Can you actually make this statement? You do not show any impact on surface concentrations. I do not think this paragraph adds any value to the manuscript and would remove.

The evaluation against surface concentration measurements does suggest an improvement in model performance with the newer emission inventories, but similar performance in the other sensitivity tests. However, as this is first covered in the following section, we see that this paragraph seems a bit out of context here and we have removed it.

Line 555: Could the higher correlation and lower bias found in the 1x1RES test not just be a consequence of improved spatial sampling in your comparison and not actually due to an improved distribution of AOD? As you do not evaluate the species whose emissions would depend more strongly on resolution (ie: dust, sea salt) it's hard to quantify the benefit here.

This is a good point, and based on previous studies of the impact of sampling and the coarseness of typical model horizontal grids on comparisons with (in particular AOD), we would expect an improved sampling to play a role here. This point is now made in the manuscript:

*"For both observables, the improvement in the 1x1RES simulation may result from a better sampling at a finer resolution, improved spatial distribution or a combination. "*

Line 638: OsloCTM3fast - what is this?? There is no prior mention of a fast configuration!!

Thanks for pointing this out, it is an error and has been corrected.

Line 688: "These results emphasize the importance of assumptions related to the BC absorption" –> these results emphasize the importance of assumptions and uncertainties related to the BC absorption.

Corrected.

Figure 6: It would be more informative to show obs +/- 1 standard deviation instead of just + 1 std.
After trying different versions, we have rather added the 25/75 percentiles as a shading to show spread in both directions.

Table S2: Column 2 you use MNB instead of NMB
Corrected, this is a typo and should be NMB here as well.

---

## Author Comment (AC3) · 11 Oct 2018

Thank you for making us aware of the missing code availability section. We have added the information on how to access the code in the revised manuscript.

---

## Author Response (AR2)

**Dear Fiona,**

Thank you for the positive feedback and comments to our manuscript. We have made additional clarifications in the methodology section that we hope address any remaining issues and misunderstandings.

Specifically, we have we have added "offline" when describing the OsloCTM3 (first paragraph section 2.1) and specified "fixed meterological data" in section 2.3. We have also modified the beginning of section 2.4, hich now reads: "We calculate the instantaneous top-of-the atmosphere radiative forcing of anthropogenic aerosols due to aerosol-radiation interactions (RFari) [*Myhre et al.*, 2013b]). The radiative transfer calculations are performed offline with a multi-stream model using the discrete ordinate method [*Stamnes et al.*, 1988]."

Thank you again for considering our study for publication in Geoscientific Model Development.

Sincerely, Marianne T. Lund CICERO, Center for International Climate Research

[revised manuscript text omitted]

Figures

---

## Author Response (AR3)

Note to editor/journal;

We have made part of the model and measurement data used in the analysis publicly available via the ACTRiS Data Center with a corresponding DOI. This DOI has been added to the "Data availability" section, along with an acknowledgement of the person assisting with this process. Otherwise, the manuscript is unchanged from the accepted version.